# Endothelial leakiness elicited by amyloid protein aggregation

Yuhuan Li [1,2,9], Nengyi Ni [3,9], Myeongsang Lee [4], Wei Wei [5], Nicholas Andrikopoulos [2,6], Aleksandr Kakinen [7], Thomas P. Davis [7], Yang Song [8] ✉, Feng Ding [4] ✉, David Tai Leong [3] ✉ & Pu Chun Ke [2,6] ✉

Alzheimer's disease (AD) is a major cause of dementia debilitating the global ageing population. Current understanding of the AD pathophysiology implicates the aggregation of amyloid beta (Aβ) as causative to neurodegeneration, with tauopathies, apolipoprotein E and neuroinflammation considered as other major culprits. Curiously, vascular endothelial barrier dysfunction is strongly associated with Aβ deposition and 80-90% AD subjects also experience cerebral amyloid angiopathy. Here we show amyloid protein-induced endothelial leakiness (APEL) in human microvascular endothelial monolayers as well as in mouse cerebral vasculature. Using signaling pathway assays and discrete molecular dynamics, we revealed that the angiopathy first arose from a disruption to vascular endothelial (VE)-cadherin junctions exposed to the nanoparticulates of Aβ oligomers and seeds, preceding the earlier implicated proinflammatory and pro-oxidative stressors to endothelial leakiness. These findings were analogous to nanomaterials-induced endothelial leakiness (NanoEL), a major phenomenon in nanomedicine depicting the paracellular transport of anionic inorganic nanoparticles in the vasculature. As APEL also occurred in vitro with the oligomers and seeds of alpha synuclein, this study proposes a paradigm for elucidating the vascular permeation, systemic spread, and cross-seeding of amyloid proteins that underlie the pathogeneses of AD and Parkinson's disease.

Alzheimer's disease (AD) is a primary form of neurological disorder entailing severe adverse effects on the memory, cognition, and life expectancy of the global ageing population. The most influential paradigm concerning AD pathology is the amyloid cascade hypothesis proposed by Hardy and Higgins in 1992[1] and its modifications thereafter, where amyloid beta (Aβ) evolves from disordered monomers to toxic oligomers and amyloid fibrils through molecular self-assembly, modulated by environmental factors such as pH, temperature, metals, chaperones, and cell membranes[2]. Accordingly, much effort over the past three decades has been made towards inhibiting or

[1]Liver Cancer Institute, Zhongshan Hospital, Key Laboratory of Carcinogenesis and Cancer Invasion, Ministry of Education, Fudan University, Shanghai 200032, China. [2]Drug Delivery, Disposition and Dynamics, Monash Institute of Pharmaceutical Sciences, Monash University, 381 Royal Parade, Parkville, VIC 3052, Australia. [3]National University of Singapore, Department of Chemical and Biomolecular Engineering, 4 Engineering Drive 4, Singapore 117585, Singapore. [4]Department of Physics and Astronomy, Clemson University, Clemson, SC 29634, USA. [5]College of Veterinary Medicine, Southwest University, Chongqing 402460, China. [6]The Nanomedicine Center, The Great Bay Area National Institute for Nanotechnology Innovation, 136 Kaiyuan Avenue, Guangzhou 510700, China. [7]Australian Institute for Bioengineering and Nanotechnology, The University of Queensland, Brisbane, QLD 4072, Australia. [8]State Key Laboratory of Environmental Chemistry and Ecotoxicology, Research Center for Eco-Environmental Sciences, Chinese Academy of Sciences, Beijing 100085, China. [9]These authors contributed equally: Yuhuan Li, Nengyi Ni. ✉e-mail: yangsong@rcees.ac.cn; fding@clemson.edu; cheltwd@nus.edu.sg; pu-chun.ke@monash.edu

clearing the toxic Aβ aggregates, employing small molecules, peptidomimetics, antibodies and, more recently, nanoparticles[3–12]. A lack of clinical success, however, has shrouded these efforts, suggesting the pathophysiology of AD is multifactorial as its triggers[13].

Indeed, it has now been realized that, in addition to Aβ amyloidogenesis, tauopathies, apolipoprotein E, and neuroimmune activation are all causative to neurodegeneration in AD[13]. The great (80–90%) correlation between AD subjects and patients carrying cerebral amyloid angiopathy (CAA) further suggests an important role of endothelial integrity in the development of AD pathogenesis[14], also evidenced by observations of cerebral endothelial dysfunction and microvascular injury induced by Aβ[15–17]. Intriguingly, while Aβ originates from the proteolytic cleavage of amyloid precursor protein (APP) in endosomal membrane[18], deposits of Aβ are seen throughout the central nervous system, cerebral blood vessels, cerebrospinal fluid, and the plasma[14,19,20]. Aβ-mediated vasoactivity, vascular capillary constriction, blood flow reduction, and paracellular transport have been reported with endothelial monolayers, blood-brain barrier (BBB), and biopsied human and rodent brain tissues[15,17,21–23], in connection with the production of reactive oxygen species (ROS), modified cytoskeletal network, altered tight-junction protein expression, and signaling to pericytes.

In this study, we report on amyloid proteins-induced endothelial leakiness (APEL) in human microvascular endothelial cell (HMVEC) monolayers resulting from their exposure to the nanoparticulates of oligomers, protofibrils, and sonicated amyloid seeds of $Aβ_{42}$ (abbreviated as "seeds" and "Aβ" hereafter), Parkinson's alpha-synuclein (abbreviated as "αS" hereafter), and FapC, a major protein constituent of the extracellular functional amyloid matrix of *Pseudomonas aeruginosa*. Endothelial leakiness was also detected in vivo with mouse cerebral vasculature exposed to Aβ oligomers and seeds. Using signaling pathway assays and steered discrete molecular dynamics (sDMD) simulations, we revealed that APEL arose from the anionic protein nanoparticulates interacting with vascular endothelial (VE)-cadherins and their associated adherens junction machinery. Our in vitro data showed that APEL originated from non-ROS and non-apoptotic events, where Aβ oligomers and seeds instead underwent direct extracellular interactions with VE-cadherins to trigger molecular pathways yielding intercellular gap formation. Through the sDMD simulations, we found that the oligomers remarkably reduced the cadherin dimer stability followed by the amyloid seeds, while flexible amyloid monomers did not affect the cadherin integrity. These results converged on a general phenomenon that anionic nanoparticulates of proteins and peptides, rendered through molecular self-assembly and fragmentation, can remodel the vascular endothelium preceding the onset of ROS, inflammation, cytotoxicity, and cerebral blood flow constriction as suggested by earlier etiological paradigms[16,21,23–26].

Phenomenologically as well as mechanistically, APEL is analogous to nanomaterials-induced endothelial leakiness (NanoEL)[27–29], whereas in the latter, anionic inorganic nanoparticles of <100 nm in size open up the endothelial paracellular route by disrupting the VE-cadherin junctions of apposing cells. The observed occurrence of APEL here was highly surprising in that proteins and peptides are biomolecules differing significantly from inorganic nanoparticles in origin, structure and function, and the density of proteins/peptides (-1.35 g/m³) is markedly below the density threshold determined for NanoEL-competent inorganic nanoparticles (1.72 g/m³)[30]. Regardless, the findings of APEL entailed rich implications, from systemic spread of amyloid proteins to their cross-seeding, BBB translocation, and clearance that underpin the contentious pathogeneses of AD and Parkinson's disease (PD).

## Results

### Characterization of amyloid protein aggregates
In our previous works[27,31,32], we found that certain nanoscale structures could interact with the VE-cadherin proteins in adherens junctions,

leading to a cascade of intracellular signaling that included actin rearrangements and development of transient intercellular gaps—a phenomenon we termed as NanoEL. Although our previously used nanoparticles were synthetic and not of a natural origin, the current evidence reiterated that it was their possession of certain physico-chemical traits, such as their size range[31] and anionic charge[33] that conferred them with the NanoEL ability, instead of their identity as being a certain material. Considering this, we hypothesized that biological nanoparticulates such as amyloid protein aggregates possessing an optimal size range, stiffness, and anionic charge, could potentially induce a similar phenomenon to NanoEL when interacting with the endothelium. Aβ (isoelectric point pI: 5.5)[34] oligomers and seeds were first chosen due to their relevance in vascular pathology and their significance to AD, as well as their polymorphic architectures. Three other types of amyloid proteins were also employed to facilitate a comprehensive understanding of APEL with respect to protein composition, size, and charge. αS, an anionic neuronal protein (pI: 4.67)[35] implicated in the pathology of PD[36,37], islet amyloid polypeptide (IAPP), a cationic human pancreatic polypeptide (pI: 8.8)[38] implicated in the pathology of type 2 diabetes (T2D)[39], and FapC (zeta potential of FapC fibrils: −36 mV)[40], a protein constituent of the *Pseudomonas aeruginosa* functional amyloid[41], were used in our study in addition to Aβ (Fig. 1a).

Fibrillization of the four amyloid protein species was performed (described within the Methods) and a thioflavin T (ThT) fluorescence kinetic assay was conducted simultaneously to monitor the amyloid formation processes. Samples at various time points were analyzed via transmission electron microscopy (TEM) to confirm their size, morphology, and structural evolution. In Fig. 1b, TEM images depicted 5 distinctly different structures of Aβ, obtained at 0 h ($Aβ_m$), 5 h ($Aβ_o$), 8 h ($Aβ_{o-p}$), and 24 h ($Aβ_f$ and $Aβ_s$), respectively, where $Aβ_f$ denoted mature fibrils and $Aβ_s$ sonicated fibril fragments. $Aβ_{o-p}$, ranged -40–100 nm in length, represents the early-stage protofibrils[42] of Aβ transitioning from the oligomers (-10–40 nm in length, Fig. 1f). Similarly, the monomers, oligomers, mature fibrils, and sonicated seeds for αS and IAPP as well as the seeds for FapC were imaged (Fig. 1c–e). Subsequently, different aggregation structures of each amyloid protein were prepared according to the time points of 0 h ($αS_m$), 20 h ($αS_o$), and 96 h ($αS_f$ and $αS_s$) for αS, 0 h ($IAPP_m$), 1 h ($IAPP_o$) and 24 h ($IAPP_f$ and $IAPP_s$) for IAPP, and 180 h for $FapC_s$ according to their ThT assays (Fig. 1f). In Fig. 1g, the sizes of the various amyloid protein aggregates were tabulated from the TEM analysis, where the oligomers of the four amyloid proteins ranged -10–70 nm in length and their seeds -20–90 nm in length, with their thicknesses below 20 nm. Note in our study, for practicality, the structurally diverse oligomers and protofibrils[42,43] were assigned based on the incubation time (by age and the ThT assay) and morphology of the peptide/protein aggregates (by TEM imaging).

### APEL was observed with anionic amyloid protein aggregates
Subsequently, we investigated the occurrence of endothelial leakiness when HMVECs were treated for 30 min with Aβ (monomers $Aβ_m$, oligomers $Aβ_o$, protofibrils $Aβ_{o-p}$, fibrils $Aβ_f$ and sonicated seeds $Aβ_s$), αS (monomers $αS_m$, oligomers $αS_o$ and seeds $αS_s$), IAPP (monomers $IAPP_m$, oligomers $IAPP_o$ and seeds $IAPP_s$), and FapC (seeds $FapC_s$). The selected µM-range protein concentrations followed the literature concerning amyloidogenesis in vitro[2]. Confocal fluorescence microscopy revealed APEL occurring in the endothelial cell monolayers exposed to the protein nanoparticulates (Fig. 2a, b). Notably, IAPP, the only positively charged peptide, did not induce any leakiness in endothelial barrier across all its four forms, including the seeds, which led to leakiness for all other three types of amyloid protein aggregates. This suggested that the characteristic of negative charge, as opposed to positive charge, is a necessary factor for the APEL phenomenon to occur. Within the two anionic pathogenic amyloid species of Aβ and αS

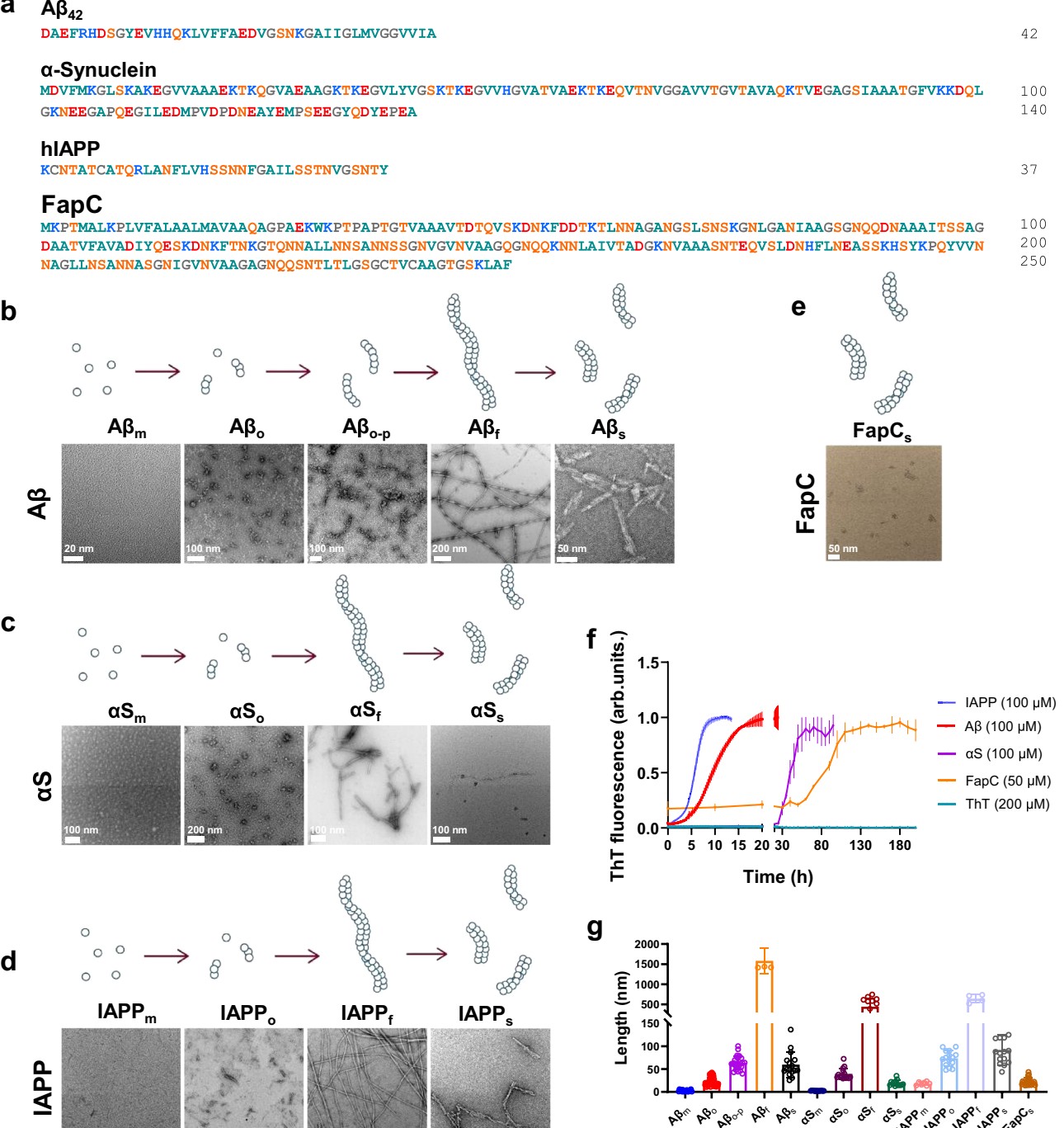

**Fig. 1 | Characterizations of aggregates of pathogenic (Aβ, αS, IAPP) and functional (FapC) amyloid proteins. a** Amino acid sequences of the four amyloid proteins used in the study. Red: negative charge, dark cyan: hydrophobic, blue: positive charge, orange: polar uncharged amino acids, and gray: special cases. **b–e** Transmission electron microscopy imaging of different amyloid protein species: **b** Aβ, **c** αS, **d** IAPP, and **e** FapC (n = 3 biologically independent samples). The label "m" stands for monomers, "o" for oligomers, "o-p" for early-stage protofibrils

that gave rise to APEL, their soluble monomers and lengthy amyloid fibrils were unable to induce leakiness.

Gap area, diameter, and frequency distribution were analyzed from the captured images (Fig. 2c and Supplementary Fig. 1, Supplementary Table 1). In Fig. 2c, for the forms of Aβ, the calculated percentage of gap area was insignificant for Aβ$_m$ when compared to the

transitioning from the oligomers, "f" for fibrils, and "s" for sonicated seeds, respectively. **f** ThT kinetic assay of amyloid protein fibrillization. Data are expressed as mean ± SD (n = 3 biologically independent samples). **g** Lengths of different amyloid protein aggregates based on TEM imaging. Data are expressed as mean ± SD (n = 17, 71, 24, 4, 16, 13, 15, 13, 14, 7, 15, 5, 4, 15 and 24 protein aggregates of Aβ$_m$, Aβ$_o$, Aβ$_{o-p}$, Aβ$_f$, Aβ$_s$, αS$_m$, αS$_o$, αS$_f$, αS$_s$, IAPP$_m$, IAPP$_o$, IAPP$_f$, IAPP$_s$, and FapC$_s$, respectively, examined over three independent samples).

non-leaky control, followed by a significant increase for Aβ$_o$, before decreasing when the aggregate length increased from Aβ$_{o-p}$ to the more rigid and lengthy Aβ$_f$, where gap area % finally became insignificant compared to control. The subsequent sonication of Aβ$_f$ to Aβ$_s$ led to a significant APEL occurrence yet again. In frequency distribution, the number of gaps induced by Aβ$_o$ was $14.3 ± 3.1 × 10^2$ gaps/mm$^2$,

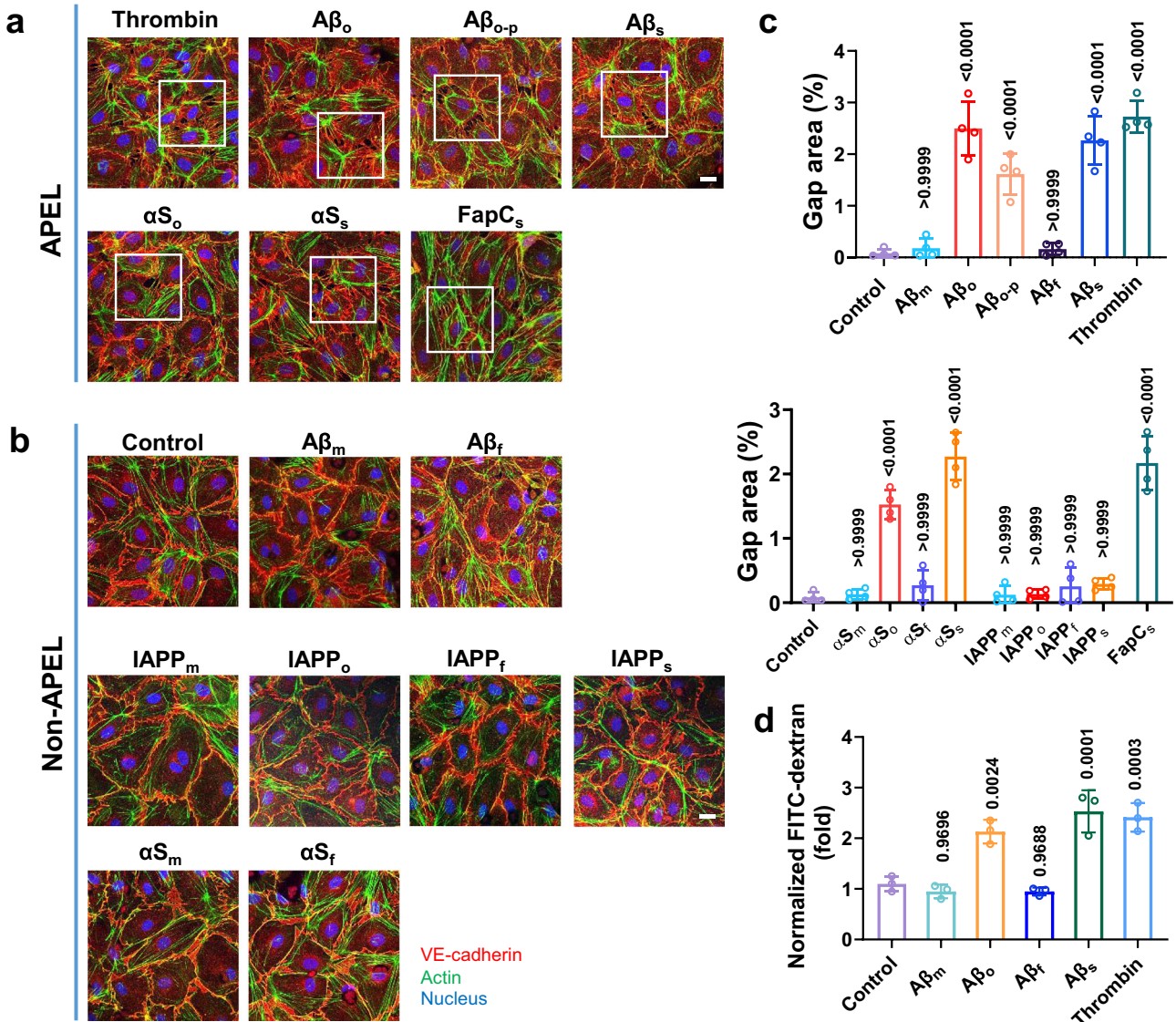

**Fig. 2 | Induced endothelial leakiness observed in HMVECs treated with anionic Aβ, αS, and FapC aggregates, but no samples of cationic IAPP induced significant leakiness. a, b** Confocal fluorescence microscopy revealed endothelial leakiness with different peptide species (Aβ 20 μM, IAPP 20 μM, αS 10 μM and FapC 1.6 μM), including Aβ oligomers (Aβ$_o$), protofibrils (Aβ$_{o-p}$) and seeds (Aβ$_s$), αS oligomers (αS$_o$) and seeds (αS$_s$), and FapC seeds (FapC$_s$), upon 30 min treatments (*n* = 3 biologically independent experiments, representative micrographs are presented). The monomers Aβ$_m$ and αS$_m$, as well as fibrils Aβ$_f$ and αS$_f$, did not induce endothelial leakiness in HMVECs. No endothelial leakiness was observed in the presence of all four forms of IAPP. Thrombin (3 U/mL) acted as positive control of induced leakiness. Scale bars: 20 μm. Red: VE-cadherin, green: actin, blue: nuclei. **c** Gap area percentages were analyzed via ImageJ according to the images related to **a, b**. Data are shown as mean ± SD (*n* = 4 images belong to three independent experiments), analyzed via one-way ANOVA with Tukey's multiple comparison tests. *P* values compared with control are shown. **d** Transwell assay quantitatively revealed occurrence of endothelial leakiness after 30 min incubation with the different forms of Aβ and other peptide species (Supplementary Fig. 2). Data are shown as mean ± SD (*n* = 3 biologically independent samples), analyzed via one-way ANOVA with Tukey's multiple comparison tests. The derived *P* values compared with control are shown.

which decreased to $9.7 \pm 3.3 \times 10^2$ gaps/mm² for Aβ$_{o-p}$ and increased back to $15.8 \pm 6.8 \times 10^2$ gaps/mm² after Aβ$_s$ were introduced (Supplementary Fig. 1). Similarly, αS$_o$ and αS$_s$ yielded significant increases in gap formation, while αS$_m$ and αS$_f$ did not elicit leakiness of significance against untreated control (Fig. 2c, Supplementary Fig. 1). Functional FapC$_s$, tested solely as seeds due to the limited relevance of their other forms in vivo, also elicited significant gap formation. Next, we employed transwell assays to quantify the extent of induced endothelial leakiness with the use of fluorescein isothiocyanate conjugated dextran (FITC-dextran) as a probe. In agreement with confocal fluorescence microscopy, the transwell assay revealed that when the amyloid protein species were incubated with HMVECs and probed for permeability changes, there were significant increases in FITC-dextran

transport across the endothelial barrier for Aβ$_o$, Aβ$_s$, αS$_o$, αS$_s$ and FapC$_s$ compared to untreated control, while Aβ$_m$, Aβ$_f$, αS$_m$, αS$_f$ and the four forms of IAPP did not yield significant results (Fig. 2d, Supplementary Fig. 2).

Collectively, APEL bore similarities to NanoEL. The differing identities of the amyloid species (such as, human or bacterial origin, pathogenic or functional) did not appear to influence the occurrence of APEL on endothelial barriers, but rather possession of relevant physicochemical properties, such as suitable size, density, and charge, was necessary. Better performances by the oligomers and sonicated Aβ seeds than fibrils, along with the short duration of induction (30 min, or likely even faster) in increasing vascular permeability were notable as APEL differed from some currently proposed mechanisms

of Aβ-induced permeability, observed typically over the periods of tens of hours or longer. These include: rat brain microvessel endothelial cells subjected to fibrillar Aβ42 for durations of 24–72 h exhibited an altered pattern of tight-junction protein expression and localization[44]; murine brain endothelial cells subjected to Aβ42 for 24 h led to increased permeability, which was proposed to result from Aβm interacting with receptors for advanced glycation end products (RAGE), leading to downstream disturbances to tight-junction protein expression and permeabilization[45]; and human umbilical vein endothelial cell barrier treated with Aβ42 for 24 h experienced downregulated junction protein expression and increased permeability[46]. Hence, we were prompted to further elucidate the specifics of the APEL process.

## In vitro APEL occurs independently of ROS generation, apoptosis, or endocytosis

We isolated one amyloid protein that resulted in significant APEL, Aβ, for subsequent characterization of the APEL process. The two forms of Aβ that induced the most significant fold changes in leakiness were Aβo and Aβs (Fig. 2d). Aβ-induced APEL was characterized through a transwell assay, where a range of concentrations (0 μM–40 μM) were employed, before two concentrations, 20 μM and 40 μM, were selected for further experiments due to their highly significant induction of

leakiness (Supplementary Fig. 3). Specifically, in the transwell assay, we measured an increase in FITC-dextran transport (and therefore, leakiness) with increasing concentrations of both Aβo and Aβs at 20 μM and 40 μM for 30 min (Fig. 3a). Next, the APEL-relevant concentrations of Aβ were employed in the examination of common toxicity effects that could lead to intercellular gap formation, which were important due to Aβ's established roles in neurotoxicity when interacting with brain endothelial cells[1]. ROS is known to lead to cell shrinkage through apoptosis[47], and free radical oxidative stress is a critical pathological effect by Aβ[48,49]. We detected no significant increase in ROS production under the different Aβ treatments for as late as 2 h, beyond the timepoint (30 min) when APEL had occurred (Fig. 3b). In complement, as seen in Fig. 3c, prior treatment with ROS scavenger N-acetyl cysteine (NAC) did not significantly reduce the degree of Aβ-induced APEL despite ROS scavengers being reported as protective against Aβ-induced oxidative stress[50]. Concurrently, a similar set of groups instead involving pre-treatment with NAC was employed. Pretreating with NAC did not decrease the overall APEL extent (Supplementary Fig. 4), based on our findings in Fig. 3b, c. A comparison between control, NAC-only, and NAC+Aβs groups affirmed that oxidative stress played no role in inducing APEL and, by deductive elimination, the APEL effect was exerted by Aβs. Gap frequency distribution analysis (Supplementary Fig. 4) supported similar overall conclusions. As

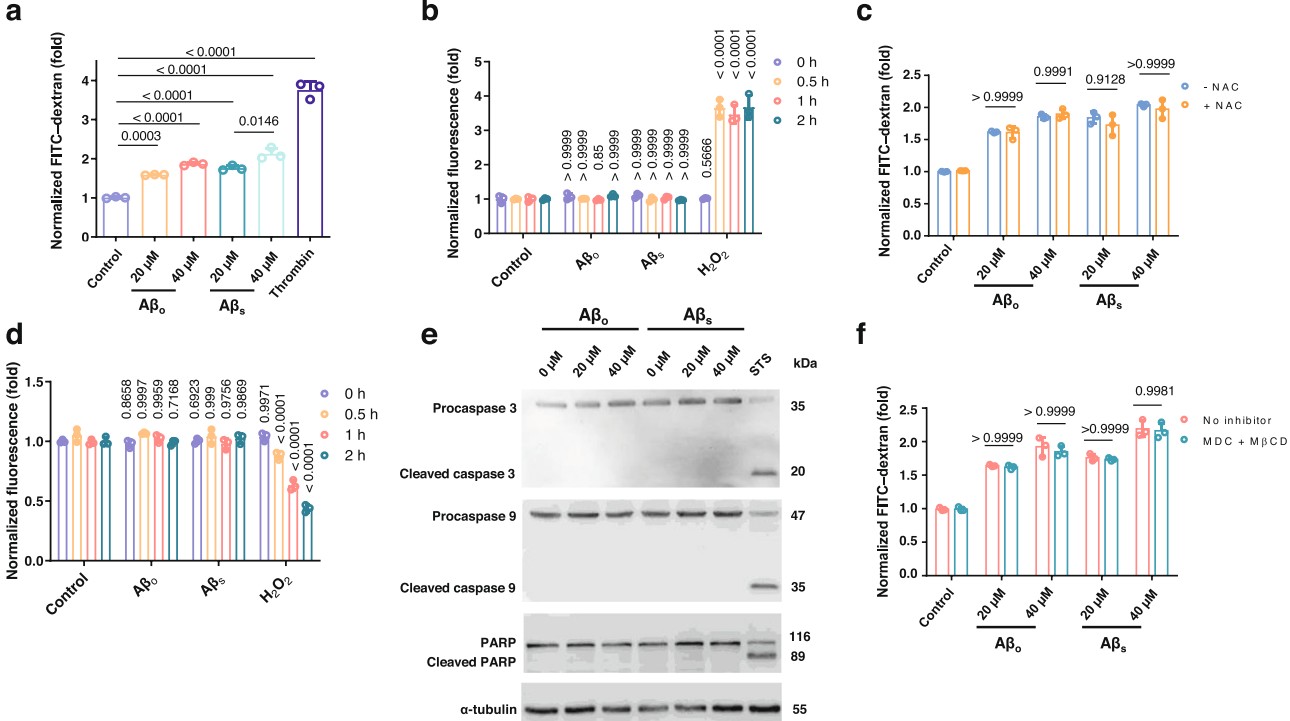

**Fig. 3 | Aβ-induced APEL is independent of increase in ROS production, decrease in cell viability, apoptotic activation, or endocytosis. a** Treatment of HMVECs with Aβo and Aβs (30 min) revealed increases in leakiness with increased concentration. Thrombin (3 U/mL) acted as positive control of induced leakiness. Results are shown as mean ± SD (n = 3 biologically independent samples), analyzed via two-way ANOVA with Tukey's multiple comparison tests. P values when compared with control are indicated. **b** No significant increase in ROS production was detected in HMVECs after treatment with either Aβ (20 μM) for as late as 2 h. Hydrogen peroxide (H₂O₂, 200 μM) acted as positive control. Results are presented as mean ± SD (n = 3 biologically independent samples) and analyzed via one-way ANOVA with Tukey's multiple comparison tests. P values for comparisons with respective control of same duration are shown. **c** Induced leakiness under Aβ was not significantly reduced with prior treatment of ROS scavenger NAC (10 mM, 1 h). Results are presented as mean ± SD (n = 3 biologically independent samples) and

analyzed via two-tailed Student's t tests. P values for comparisons with respective non-NAC-treated groups are shown. **d** No decline in HMVEC viability was detected after treatment with different Aβ proteins (20 μM) for as late as 2 h. Results are presented as mean ± SD (n = 3 biologically independent samples) and analyzed via one-way ANOVA with Tukey's multiple comparison tests. P values compared with respective control of same time duration are shown. **e** Immunoblotting revealed no activation of apoptotic markers within concentrations used for Aβ APEL. Staurosporine (STS; 1 μM, 2 h) served as positive control for apoptosis. A representative blot is presented, out of n = 3 biologically independent experiments. **f** Inhibition of endocytosis in HMVECs (mixture of 5 mM MβCD and 10 μM MDC) did not result in a significant reduction in Aβ APEL. Data are presented as mean ± SD (n = 3 biologically independent samples) and analyzed via two-tailed Student's t tests. P values compared with respective non-inhibitors-treated groups are shown.

anticipated, NAC+ Aβ$_s$ group ($18.7 \pm 5.3 \times 10^2$ gaps/mm$^2$) yielded no significant decrease in gap frequency compared to the Aβ$_s$-only group ($21.7 \pm 2.4 \times 10^2$ gaps/mm$^2$).

In addition, apoptosis, a known contributor to cell shrinkage[51], was also reported as a toxic effect induced by Aβ structures from oligomers to fibrils, involving caspases such as 3, 8, and 9[52–54]. Firstly, no decline in cell viability was detected under Aβ-induced APEL conditions (Fig. 3d). Subsequently, as shown in Fig. 3e, we detected no activated forms of important apoptotic enzymes, namely caspase 9 (an initiator caspase), caspase 3 (an effector caspase), and poly(ADP-ribose) polymerase (PARP; a downstream cleavage target) during Aβ-induced APEL conditions[55,56]. Furthermore, endocytosis was found to not significantly contribute to Aβ-induced APEL as the mixture of endocytic inhibitors methyl-β-cyclodextrin (MβCD) and mono-dansylcadaverine (MDC) could not significantly reduce leakiness induced in the HMVEC barrier (Fig. 3f). Concurrently, confocal fluorescence imaging was also conducted on HMVECs treated with ThT-labeled Aβ$_s$ for the typical duration of 30 min, but with no prior endocytic inhibitors (Supplementary Fig. 5). The Z-stacked images revealed that Aβ$_s$ particles were generally present (i.e., more brightly fluorescent) on a higher plane above the plane of focus of the HMVECs, suggesting that Aβ$_s$ particles, even without endocytic inhibitors treatment, were minimally endocytosed at the duration of 30 min of exposure. Collectively, Aβ-induced APEL was found to be not significantly attributable to endothelial toxic effects typical to Aβ, most likely due to the short duration (30 min) of cell exposure to Aβ, and was triggered extracellularly, bearing strong similarities to NanoEL[57]. Similar responses to the NAC and the endocytic inhibitor treatments could also be replicated in a lower Aβ concentration of 10 μM (Supplementary Fig. 6).

### In vitro Aβ-induced APEL involves extracellular interactions with VE-cadherin

To validate the extracellular triggering of the APEL process and potential involvement of Aβ with adherens junction protein VE-cadherin, we further performed a competitive inhibition assay with Aβ$_s$. A treatment of VE-cadherin antibody blocked the extracellular EC1 or EC2-3 domain of VE-cadherin before or after their incubation with Aβ. As captured in confocal fluorescence microscopy in Fig. 4a and Supplementary Fig. 7 and then analyzed semi-quantitatively in Fig. 4b, c, the percentage of gap area formation was significantly reduced at the EC1a dilutions of 1:100 and 1:500 pre-treatment (EC1a+Aβ$_o$/Aβ$_s$) as compared to Aβ$_o$/Aβ$_s$ only. More gap areas were observed for the EC1a +Aβ$_o$/Aβ$_s$ group than the EC1a only at 1:500, suggesting the lower amount of EC1a could not fully block the effects exerted by Aβ$_o$/Aβ$_s$. Besides, there was no significant difference in gap area percentage between the EC1a dilution of 1:20 pre-treatment and Aβ$_o$/Aβ$_s$ only. As the antibody concentration increased, the VE-cadherin junctions between adjacent cells became unstable to gradually break apart to render gaps, especially at the dilution of 1:20. Therefore, it was not accurate to evaluate the blocking effect at such high antibody concentrations. Hence, at the proper concentrations of EC1a pre-treatment (1:100 and 1:500), most of the interactions between Aβ$_o$/Aβ$_s$ and VE-cadherin were blocked. On the other hand, no significant difference in the gap area formation was found between Aβ$_o$/Aβ$_s$ only and the groups of EC1a addition after incubation with Aβ$_o$/Aβ$_s$ (Aβ$_o$/Aβ$_s$ + EC1a), indicating the post-treatment EC1a failed to eliminate the effects induced by Aβ$_o$/Aβ$_s$ (Fig. 4c and Supplementary Fig. 7). Similar phenomena were observed with the incubation of Aβ$_o$/Aβ$_s$ and BV-6 antibody (Supplementary Fig. 8). Taken together, the extracellular EC1 and EC2-3 domains of VE-cadherin were not only important but essential for the occurrence of APEL induced by the amyloid aggregates.

Next, we employed a co-localization assay where cells were treated with ThT-labeled Aβ$_s$ and VE-cadherin was later labeled during immunofluorescence staining. Under confocal fluorescence microscopy, it was observed that bright yellow dots were yielded, indicating ThT-labeled Aβ$_s$ (in green channel) and high VE-cadherin expression levels (in red channel) were co-localized (Fig. 4d). The yellow dots were located along the perimeters of HMVECs, where the cell junctions were found, and most were just next to or inside the intercellular gaps. This assay further suggested that interactions took place between Aβ$_s$ and VE-cadherins during the APEL process. In addition, we employed a co-immunoprecipitation assay with the four forms of Aβ to further support the co-localization assay's findings (Fig. 4e). We observed the presence of interactions between VE-cadherin and the species Aβ$_o$ and Aβ$_s$, where the pulldown of VE-cadherin also increased with higher Aβ treatment concentrations. However, no detectable interactions were found between VE-cadherin and the two species of Aβ$_m$ and Aβ$_f$. This experiment further supported the occurrence of external interactions between VE-cadherin proteins and the eligible Aβ nanoparticulates (Aβ$_o$ and Aβ$_s$) during APEL, while Aβ$_m$ and Aβ$_f$ did not elicit such interactions likely due to their softness (for Aβ$_m$) or large length and rigidity (for Aβ$_f$) for entering the highly confined adherens junction.

### In vitro Aβ-induced APEL involves VE-cadherin signaling and actin remodeling

To further elucidate the molecular mechanisms for Aβ-induced APEL, we examined the effects of APEL on the VE-cadherin signaling pathway. Canonically, VE-cadherin signaling includes phosphorylation at two important residues tyrosine 658 (Y658) and tyrosine 731 (Y731), which leads to downstream reduced interactions with p120 and β-catenin, with actin rearrangement and appearance of intercellular retractions as notable endpoints[58–61]. Our previous works with NanoEL revealed that a similar cascade could also be triggered by synthetic nanoparticles with certain physicochemical parameters, despite their exogenous origin[31,57]. In a similar manner, we observed increased phosphorylation of Y658 and Y731 residues of VE-cadherin under Aβ$_o$ treatment, where the degree of phosphorylation increased with the concentration of Aβ (Fig. 5a). Src-kinase inhibitor, PP1, which inhibits the kinase responsible for phosphorylation of the residues, was also employed as a pre-treatment, where it was observed that the degree of phosphorylation under every Aβ$_o$ treatment decreased compared to their relevant control but was incompletely inhibited (Fig. 5a–c). Likewise, Aβ$_s$ treatment resulted in increased phosphorylation of the two residues, and application of PP1 pre-treatment modulated an increase in phosphorylation (Fig. 5d–f). Complementarily, in our parallel transwell assay, PP1 pre-treatment also resulted in a significant reduction of induced leakiness under the treatment of either of the Aβ species, when compared to their non-PP1 pre-treated counterparts (Fig. 5e). This further evidenced the involvement of activated VE-cadherin signaling during the APEL phenomenon. To validate the involvement of actin remodeling in the APEL event, we also employed a Rho-associated, coiled-coil containing protein kinase (ROCK) inhibitor, Y27632, which disrupts normal actin functioning through destabilizing focal adhesions and stress fibers[62]. The transwell assay revealed that pre-treatment with Y27632, which interfered with normal actin remodeling, led to a suppressed induction of leakiness by Aβ$_o$ or Aβ$_s$ compared to the respective untreated groups (Fig. 5f). Similar responses to PP1 and Y27632 were also seen at a lower Aβ concentration of 10 μM (Supplementary Fig. 6). Together, these results revealed the role of VE-cadherin signaling and actin remodeling as part of the molecular mechanisms involved in APEL.

In addition, we verified that during the time frame of APEL occurrence, tight-junction activities appeared to be insignificant, where key tight-junction proteins such as ZO-1 (zonula occludens protein 1), occludin and claudin-5 did not experience significant downregulation (Supplementary Fig. 9), as opposed to previously reported phenomena[45]. Furthermore, unlike for VE-cadherin, co-immunoprecipitation assays revealed no detectable interactions

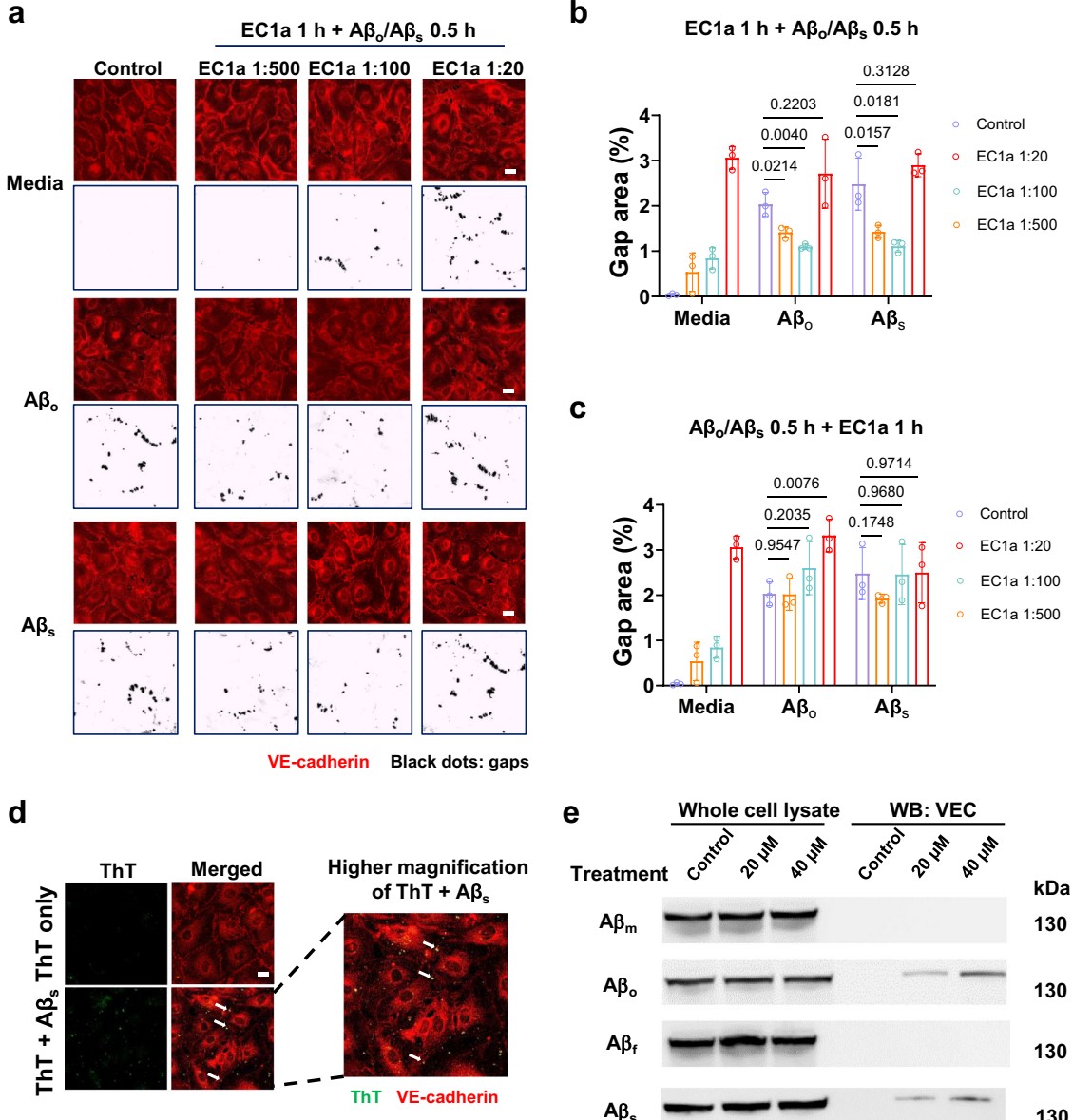

**Fig. 4 | Extracellular interactions with VE-cadherin are necessary for APEL induction by Aβ aggregates. a** APEL arising from Aβ_o or Aβ_s (20 μM, 30 min) was inhibited by an EC1 domain antibody (EC1a) pre-treatment (EC1a dilutions 1:100 and 1:500, 1 h prior to amyloid protein treatments), which specifically blocked the extracellular domain of VE-cadherin ($n = 3$ biologically independent experiments, representative micrographs are presented). The post-addition of EC1a after 0.5 h-incubation of amyloid proteins did not negate the EL occurrence induced by Aβ_o/Aβ_s (Supplementary Fig. 7). Gap distributions were obtained from captured confocal images through trainable Weka segmentation plugin in ImageJ software. Scale bars: 20 μm. Black dots in the images represent holes in the HMVECs monolayer. Red: VE-cadherin. Y and X axes revealed the combination of pre-treatment (Media/EC1a) and Aβ_o/Aβ_s (−/+) employed. **b**, **c** Gap area percentages were analyzed by

ImageJ according to the images from panel **a** and Supplementary Fig. 7. Data are shown as mean ± SD ($n = 3$ images belongs to three biologically independent samples), analyzed via two-tailed Student's $t$ tests. The derived $P$ values compared between groups are shown. **d** Co-localization of ThT-labeled Aβ_s and VE-cadherin was observed in HMVECs near their cell junctions, revealing association between them ($n = 3$ biologically independent experiments). Red: VE-cadherin, green: ThT-labeled Aβ_s. Scale bars: 20 μm. **e** Co-immunoprecipitation assay pulled down with an anti-APP antibody, for probing the interactions between VE-cadherin (VEC) and the four forms of Aβ following HMVECs exposure to Aβ_o (0, 20, and 40 μM, 30 min). A representative blot is displayed, out of three biologically independent experiments.

between the APEL-eligible Aβ_o and tight-junction proteins such as occludin and claudin-5. These results did not support the involvement of tight-junction activities with Aβ_o during the APEL time frame (Supplementary Fig. 10).

**Aβ-induced APEL in in vivo BBB models**

Human brain endothelial cells hCMEC/D3 were employed as a BBB model in the transwell assay format and exposed to 0–40 μM of Aβ_s. An increase in the penetration of FITC-dextran was observed with the

increasing concentration of Aβ_s, with 40 μM of Aβ_s attaining the highest permeability compared to control of 0 μM, while 5 μM yielded a comparatively insignificant difference (Supplementary Fig. 11). Considering the significant induction of permeability by Aβ species in the in vitro BBB model, we further determined the effects of Aβ in vivo with APP/PS1 mice, a transgenic AD model using mice of two different ages (2 months to represent young mice before the onset of AD and 12 months to represent aged mice with the onset of AD). The concentrations of Aβ_{42} in the blood and tissues of APP/PS1 mice and the

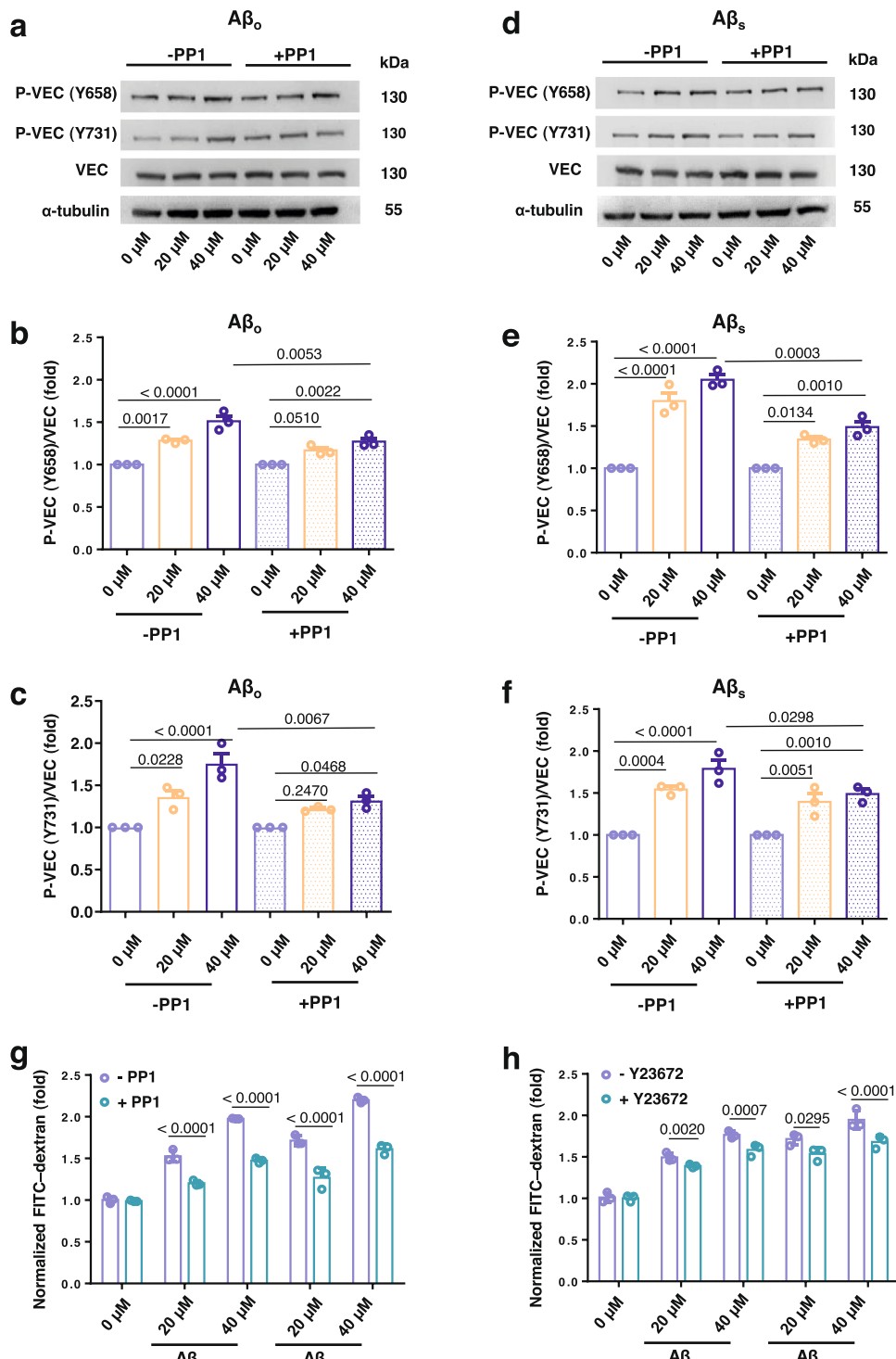

**Fig. 5 | Aβ-induced APEL involves VE-cadherin signaling and actin remodeling.**
**a–c** Aβ$_o$ treatment on HMVECs (20 μM, 40 μM) induced activation of VE-cadherin (VEC) signaling, including its phosphorylation at tyrosine residues of 658 (P-VEC(Y658)) and 731 (P-VEC(Y731)). Pre-treatment with Src-kinase inhibitor, PP1 (10 μM, 1 h), led to attenuated activation of these residues. Results presented are shown as mean ± SD ($n = 3$ biologically independent samples). **d–f** Similarly, immunoblotting, and corresponding semiquantitative analysis of Aβ$_s$ treatment revealed activation of VEC signaling, with reduced activation under PP1 pre-treatment. Data are mean ± SD, $n = 3$ (biologically independent samples,

representative blots presented here), analyzed via one-way ANOVA with Tukey's multiple comparison tests. $P$ values from respective stated comparisons are shown. **g** Transwell assay affirmed the significant reduction of APEL (Aβ$_o$ or Aβ$_s$, 30 min) due to pre-treatment with PP1 (10 μM, 1 h), when compared to respective counterparts without PP1. **h** Aβ APEL involved actin remodeling. Pre-treatment with a RhoA kinase (ROCK) inhibitor, Y27632 (10 μM, 1 h) led to significantly decreased leakiness under Aβ$_o$ and Aβ$_s$ (30 min). Results presented are shown as mean ± SD ($n = 3$ biologically independent samples), analyzed via two-tailed Student's $t$ tests. $P$ values between compared groups are shown.

background strain mice (C57BL/6 J) at 2- and 12 months old were measured and compared using a mouse $A\beta_{42}$ ELISA kit. The $A\beta_{42}$ expression in the brain of APP/PS1 mice at 12 months old was almost twice as high as in young AD mice at 2 months old. Not only in the brain, but also in the blood and the liver, we observed significant differences in the $A\beta_{42}$ expression between the 2- and 12-month-old mice. The $A\beta_{42}$ expression in C57BL/6 J kept largely unchanged in the 2- and 12-month-old mice (Supplementary Fig. 12). Furthermore, the APP/PS1 mice were subjected to intravenous injection of Evans Blue (EBD) solution. After 24 h, we observed that EBD permeabilization occurred in both the 2- and 12 months APP/PS1 mice (Fig. 6a). As expected, the leakiness of EBD in the brain of the 12 months old mice was significantly enhanced ($P < 0.05$) (Fig. 6b), accompanied by significant increases of leaked EBD in the liver and the lungs (Supplementary Fig. 13). The leakiness of EBD in the brain of background strain C57BL/6 J mice remained unchanged at 2- and 12 months old (Supplementary Fig. 14).

We further studied endothelial leakiness induced by $A\beta$ in Swiss mice, where $A\beta_s$ or $A\beta_o$ ranging from 0.002 µg to 100 µg per mouse (Fig. 6c, d, Supplementary Figs. 15–18) or solely EBD was intravenously injected. Fluorescence imaging of mice brains post-sacrifice revealed a markedly increased permeability across the BBB for both $A\beta_s$ and $A\beta_o$, even at the low concentration of 0.02 µg per mouse, compared to EBD-only treatment (Fig. 6c, d). The fluorescence intensity of leaked EBD increased significantly with the increasing dosage of both $A\beta_s$ and $A\beta_o$. In the lower dosage range from 0.002 to 2 µg per mouse, the intensity of leaked EBD in the brain was ~1.1-1.4 times that of control, and when the dose of $A\beta$ was above 20 µg per mouse, the fluorescence intensity reached 4.2-5.2 times that of control (Supplementary Fig. 17a, b). Meanwhile, we observed elevated fluorescence intensities of leaked EBD in other tissues, including in the heart, the spleen, and the diaphragm (Supplementary Figs. 16, 18). Furthermore, transverse brain sections were also derived from mice after similar injected treatments for an immunofluorescence analysis after staining with an anti-$A\beta$ antibody (Fig. 6e, Supplementary Fig. 17c). The presence of $A\beta$ structures resembling seeds and oligomers was found in both the $A\beta_s$ and $A\beta_o$ groups, as inferred from the anti-$A\beta$ antibody staining. The red clusters of $A\beta$ wrapped in gold EBD indicated locations of blood vessels, while gold EBD diffusing in the tissues surrounding the blood vessels revealed endothelial leakiness. Together, the in vivo findings revealed an increased permeability in the brains of mice after treatment with $A\beta$ species, as corroborated by a concurrent presence of $A\beta$ structures within the brains.

Apart from EBD, we also employed FITC-dextran (10, 000 Da) as another permeabilization indicator to confirm endothelial leakiness induced by the $A\beta$ aggregates of 2 and 20 µg at 3 and 24 h in Swiss mice (Supplementary Fig. 19). There was no significant APEL observed in vivo at 3 h administration with 2 and 20 µg of $A\beta_o/A\beta_s$. With the observation time extended to 24 h, APEL occurred in the brain under the treatment of $A\beta_o/A\beta_s$. In contrast, no marked vascular leakiness was observed at 3 h and 24 h with the administration of $IAPP_s$ (Supplementary Fig. 20).

In addition, the reversibility of in vivo vascular leakiness induced by $A\beta$ species was tested. Significant inhibition of APEL was observed at the injection of PP1 after 3 h-administration of $A\beta_s$ ($A\beta_s$ (3 h) + PP1) of 2 and 20 µg, compared with $A\beta_s$ only, suggesting that vascular leakiness induced by $A\beta_s$ was through the VE-cadherin signaling pathway and the APEL phenomenon could be reversed in vivo (Fig. 6f, g). This result was consistent with the in vitro data in Fig. 5.

## $A\beta$ interactions with VE-cadherin dimers characterized in silico

To understand the disruption of VE-cadherin dimer induced by amyloid proteins at the molecular level, we employed all-atom discrete molecular dynamics (DMD) and steered DMD (sDMD) simulations. We first performed binding simulations of VE-cadherin with $A\beta$ species.

Specifically, we employed a EC1 cadherin dimer from the full length of a VE-cadherin dimer to efficiently mimic the *trans* interactions (Fig. 7a). From our recent study, we successfully demonstrated that the EC1 dimer was a suitable model to evaluate the dimer stability in the presence and absence of gold nanoparticles (AuNPs)[57]. Next, three different forms of $A\beta$ monomer ($A\beta_m$), oligomer ($A\beta_o$), and tetramer seed ($A\beta_s$) were prepared to assess their interactions with the cadherin dimer and how the binding disrupted the dimer (Fig. 7b). The three different forms of $A\beta$ were randomly located near the EC1 dimer, and 40 independent binding DMD simulations for 50 ns were performed. For the binding simulations, we computed the binding frequencies of the peptide species with the EC1 dimer. We observed that each form of $A\beta$ entailed distinct binding behaviors with the EC1 dimer (Fig. 7c). Our recent study revealed that an AuNP coated with citric acids preferred to bind the turn region of the dimer mostly consisted of cationic amino acids[57]. Similarly, we observed that the $A\beta_o$ highly bound to the turn regions of the EC1 dimer. However, the $A\beta_s$ mostly bound to the C-terminus of the dimer while the $A\beta_m$ bound to the entire region of the dimer. The colored binding frequencies on the surface of the EC1 dimer detailed the binding behaviors of the $A\beta$ species with the EC1 dimer (Fig. 7d). Such different binding behaviors can also be verified by computing the binding frequency of the EC1 dimer with the $A\beta$ species (Supplementary Fig. 21a). Specifically, the first 10 residues of $A\beta_o$ and middle 15 residues (15–30) of $A\beta_s$ mostly interacted with the EC1 dimer, where all amino acids of the $A\beta_m$ highly bound to the EC1 dimer. In addition, hydrophobic-residue interactions between the EC1 dimer and $A\beta_m$ induced the binding, while charged-residue interactions between the EC1 dimer and $A\beta_s$ drove the binding (Supplementary Fig. 22). Except for the monomer, the preferred binding sites of the $A\beta_o$ and $A\beta_s$ with the EC1 dimer were different from each other due to the distinct conformations of the $A\beta$ nanoparticulates (Supplementary Fig. 21b).

After the binding DMD simulations, all-atom sDMD simulations were carried out to understand the EC1 dimer stability in the presence and absence of the $A\beta$ species. For the sDMD simulations, one of the EC1 domains was immobilized and the other side of the domain stayed flexible. Constant forces in the range of 0-60 pN with 10 pN of windows were applied to the flexible domain of the EC1 dimer toward the EC2 domain of the VE-cadherin dimer (Fig. 7e). Subsequently, we performed 70 independent sDMD simulations with randomized initial velocities assigned according to Maxwell-Boltzmann distribution and each sDMD simulation lasted for 100 ns (Supplementary Table 3). Then, we evaluated the dimer stability with and without $A\beta$ species by violin plots as a function of first mean dissociation time and applied forces after the sDMD simulations (Fig. 7g–i). The first mean dissociation time was determined when the number of contacts at the dimer interfaces was reduced to zero. Here, we note that the 100 ns of the dissociation time means that the EC1 dimer stayed associated during the sDMD simulations. We observed that the $A\beta_o$ significantly increased the probability of early cadherin dimer dissociation, followed by the seed. Specifically, $A\beta_o$ elicited the highest cadherin dimer dissociation under the low force range (0-30 pN). However, the effect of $A\beta_m$ on the dimer disruption was negligible for all applied forces. Representative trajectories of the $A\beta_o$- and $A\beta_s$-EC1 complexes reflected the early dissociation of the cadherin dimer (Fig. 7j, k). $A\beta_m$ was detached from the dimer at an early-stage or moved around during the sDMD simulations due to the flexible nature of the monomer (Supplementary Fig. 23). Hence, although $A\beta_m$ exerted a similar binding strength to EC1 as $A\beta_o$ or $A\beta_s$ (Fig. 7c), their different effects on cadherin dimer stability were mainly due to the greater conformational dynamics of $A\beta_m$ compared to $A\beta_o$ and $A\beta_s$ (Fig. 7j, k). To confirm the reduced cadherin dimer stability, we measured the RMSF of the flexible domain of the EC1 dimer (Fig. 7f). In a recent study[57], we showed that a reduced entropy disrupted the inherent function of a cadherin dimer. Our calculated RMSF results indicated that the $A\beta_o$ and $A\beta_s$

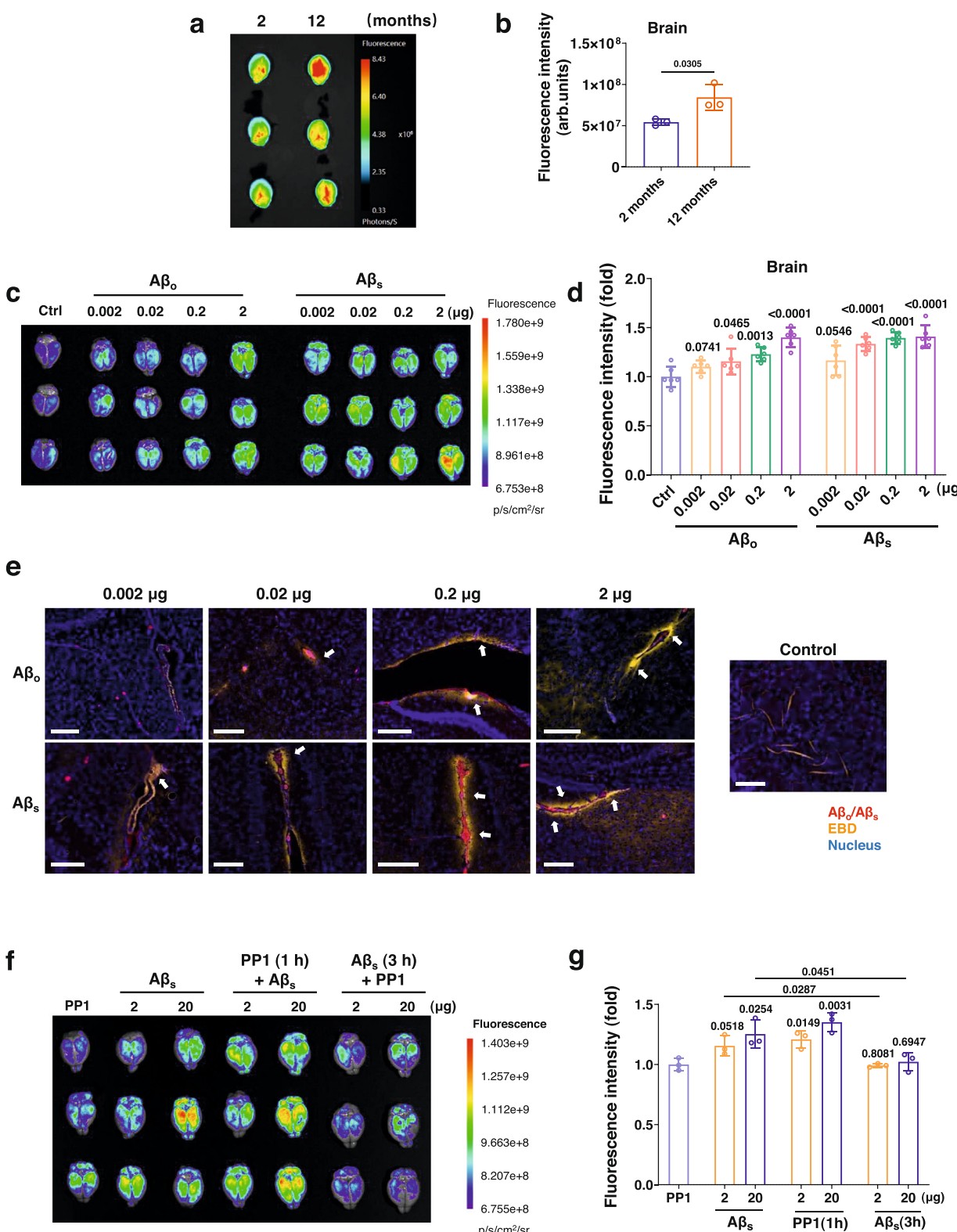

considerably reduced the overall RMSF, while $Aβ_m$ slightly reduced the RMSF for some amino acids. Although the Aβ species and the EC1 dimer in the simulations were not entirely the same sizes as in the experiments due to the high computational costs, the overall tendencies of the violin plots indicated that both $Aβ_o$ and $Aβ_s$ increased the dissociation (Supplementary Table 4) and reduced the dimer stability of VE-Cadherin. As shown in our prior study[57], a reduced dimer stability and an increased dissociation of VE-cadherin resulted in a

lower critical tensile force to induce intercellular gap stabilized by a cluster of VE-cadherins. Constantly experiencing dynamic mechanical stretches in living endothelial cell membrane, the adherens junctions with compromised VE-cadherins open and form gaps readily. Therefore, our simulation results were consistent with the experimental data (Figs. 4–6).

To test whether the Aβ species bound to other extracellular domains of the VE-cadherin dimer, we additionally performed a

**Fig. 6 | Aβ-induced vascular leakiness in vivo. a, b** In vivo leakiness comparison of APP/PS1 mice at 2- and 12 months old through measurement of EBD's permeabilization 24 h post injection. Results are shown as mean ± SD ($n$ = 3 animals), analyzed via two-tailed Student's $t$ tests. $P$ values compared with control are shown. **c, d** In vivo experiment involving injection of $Aβ_s$ or $Aβ_o$ (0.002, 0.02, 0.2, and 2 µg) into Swiss mice revealed increased leakiness across the blood-brain barrier after 24 h. Results are shown as mean ± SD ($n$ = 6 animals, including data for 3 male mice per group shown in this panel and data for 3 female mice per group shown in Supplementary Fig. 15), analyzed via two-tailed Student's $t$ tests. $P$ values compared with control are shown. **e** Immunofluorescence analysis of transverse brain sections after 24 h-injection with $Aβ_s$ and $Aβ_o$ revealed leaked EBD (indicated with white arrows) diffusing in the brain, compared to the brain of mice injected with only EBD (control) ($n$ = 3 biologically independent tissues). Red: anti-Aβ antibody (6E10),

blue: DAPI, gold: EBD, Scale bars: 200 µm. The results for injections of 20, 50, and 100 µg $Aβ_s$ or $Aβ_o$ are shown in Supplementary Figs. 17 and 18. The selected amyloid protein doses spanned over five orders of magnitude to reflect both the physiological and pathogenic AD conditions in vivo[82] and to ensure observations of the APEL phenomenon by fluorescence microscopy and complementary assays. **f, g** APEL arising from $Aβ_s$ (2 and 20 µg, 24 h) in Swiss mice was reversed by Src-kinase inhibitor, PP1. PP1 (1 h) + $Aβ_s$ refers to pre-injection of PP1 at 1 h prior to $Aβ_s$ administration, and $Aβ_s$ (3 h) + PP1 refers to 3 h administration of $Aβ_s$ followed by PP1 injection. A significant reduction of the EBD fluorescence intensity in the brain, measured after 24 h-injection of $Aβ_s$, was observed in $Aβ_s$ (3 h) + PP1 compared to $Aβ_s$ only at both 2 and 20 µg. Results are presented as mean ± SD ($n$ = 3 animals), analyzed via two-tailed Student's $t$ tests. $P$ values compared between groups are shown.

binding simulation of Aβ with an EC1-2 dimer (Supplementary Fig. 24). We found that all three Aβ species – the monomer, the oligomer, and the seed - could bind both the EC1 and EC2 domains. The Aβ-binding profiles were similar in simulations with the EC1 and EC1-2 dimers, in agreement with the APEL blocking assays with EC1a and BV-6 (Fig. 4 and Supplementary Fig. 8). In practice, the EC1 dimer could still be favored as the binding site over other EC domains due to steric constraints imposed by the VE-cadherin-cell membrane architecture. Nevertheless, as long as $Aβ_o$ and $Aβ_s$ bind the EC1 dimer, the disruption as revealed by our sDMD simulations with EC1 dimer is expected to occur. Furthermore, although only the 1:1 ratio between Aβ species and VE-cadherin was probed in silico due to limitations of computational costs, an accumulative effect of binding-induced VE-cadherin dimer disruption by multiple $Aβ_o$ and $Aβ_s$ could take place due to the availability of multiple binding sites in the dimer (e.g., Fig. 7d).

## Discussion

Molecular assembly of proteins and peptides is, in essence, a remarkable nanotechnology employed by biological systems to render functional and pathogenic outcomes. Within the framework of AD, the self-assembly of $Aβ_m$ into oligomers, protofibrils, and amyloid fibrils entails rich pathogenic implications that are not fully understood, as reflected by a lack of success in clinical trials targeting protein amyloid aggregation. While CAA and compromised BBB usually accompany the symptoms of dementia[14], the exact cause of cerebral vascular damage and their relationship with the development of AD remain unclear, with ROS production, inflammation, and the physical breakdown of tight junctions implicated as causative instigated by Aβ, Tau and apolipoprotein E, among others[14,15,17,21–23].

In this study, we report that the oligomers and seeds but not monomers or fibrils of either pathogenic (i.e., Aβ and αS) or functional amyloid proteins (i.e., FapC), ubiquitously elicited APEL in HMVECs whose characteristics were reminiscent of NanoEL, a biological phenomenon entailed by inorganic nanoparticles of certain size (<100 nm) and charge (anionic or near neutral)[29,63,64]. Specifically, our signaling pathway and transwell assays, as well as fluorescence imaging, complemented with atomistic DMD and sDMD simulations, revealed disruption to the VE-cadherin machinery introduced by $Aβ_o$ and $Aβ_s$. In contrast, $Aβ_m$, $Aβ_f$ and all forms of cationic IAPP were incompetent in inciting APEL. These findings suggested that, aside from size and charge to fit within the finite paracellular space and evade endocytosis, the stiffness of amyloid proteins, elevated in the oligomers and protofibrils than the monomers due to β-sheets stacking through fibrillization (Supplementary Figs. 25, 26), could also play a role in the occurrence of APEL.

Notably, our signaling pathway assays revealed that APEL was independent of ROS production, cytotoxicity, and endocytosis but mediated by the protein nanoparticulates engaged with the extracellular domains of VE-cadherins, triggering intracellular actin network reorganization as well as intercellular gap formation (Figs. 3–5). These latter findings were consistent with that of NanoEL but distinguished

sharply from the existing literature linking vasculature damage and AD[13,65], both in terms of the mechanisms and the spatiotemporal characteristics of intercellular gap formation. Together, this study highlighted effects of the nano-dimensionality of amyloid proteins on the endothelia as a potential contributor to vasculature damage and the etiology of AD. In light of the pervasive distributions of the human vasculature and Aβ, the mechanism revealed here may spell implications for the cross-talk between AD and other amyloid pathologies[15,17,65,66].

## Methods

### Aβ, IAPP, αS, and FapC sample preparations

1 mg of $Aβ_{42}$ ($_1$DAEFRHDSGYEVHHQKLVFFAEDVGSNKGAIIGLMVGGV VIA$_{42}$, sourced from AnaSpec Inc., purity ≥95%, molecular weight (MW): 4,514 Da) and IAPP ($_1$KCNTATCATQRLANFLVHSSNNFGAILSS TNVGSNTY$_{37}$, sourced from AnaSpec Inc., purity ≥95%, MW: 3,905 Da), were treated with 1 mL of hexafluoro-2-propanol (HFIP, Sigma-Aldrich, USA) for 3 h at room temperature to break down the pre-existing aggregates. The solutions were aliquoted and freeze-dried for future use. The dried Aβ was dissolved in 0.1% $NH_4OH$ and diluted in MilliQ $H_2O$ to a stock concentration. FapC monomers were produced and purified following the protocol described in our prior publication[67]. The dried IAPP, lyophilized αS ($_1$MDVFMKGLSKAKEGVVAAAEKTKQG VAEAAGKTKEGVLYVGSKTKEGVVHGVA  TVAEKTKEQVTNVGGAVVTG VTAVAQKTVEGAGSIAAATGFVKKDQLGKNEEGAPQEGILEDMPVDPDN- EAYEMPSEEGYQDYEPEA$_{140}$, sourced from AlexoTech, purity ≥95%, MW: 14,460 Da), and FapC were respectively dissolved in MilliQ $H_2O$.

To acquire different species of the amyloid proteins, the aqueous solutions of Aβ, IAPP, αS, and FapC were incubated at 37 °C and allowed to fibrillate for different times according to a thioflavin T (ThT, Sigma-Aldrich, USA) kinetic assay. The seeds were derived from sonicated mature fibrils of different amyloid proteins using a Vibra-Cell™ Ultrasonic VCX 750 sonicator equipped with a 3 mm microtip. 2 min sonication at 20% of the maximum output power of the sonicator was applied.

### Thioflavin T kinetic assay

For the ThT assay, 50 µL aqueous solutions of 100 µM IAPP, 100 µM Aβ, 100 µM αS, or 50 µM FapC were incubated with 200 µM of ThT respectively in a black, clear bottom 96-well plate at 37 °C. The incubation conditions were kept the same for all the amyloid proteins except for aS, where glass beads were added into each well and shaken at 200 rmp for 5 min before reading[68]. The fluorescence intensity was recorded with excitation at 440 nm and emission at 484 nm on a microplate reader (CLARIOstar, BMG LABTECH). All experiments with the samples were performed in triplicate.

### Transmission electron microscopy

TEM images of all amyloid proteins at different time points according to the ThT result were acquired using a Tecnai F20 electron microscope (200 kV). 10 µL of each sample was placed onto glow discharged

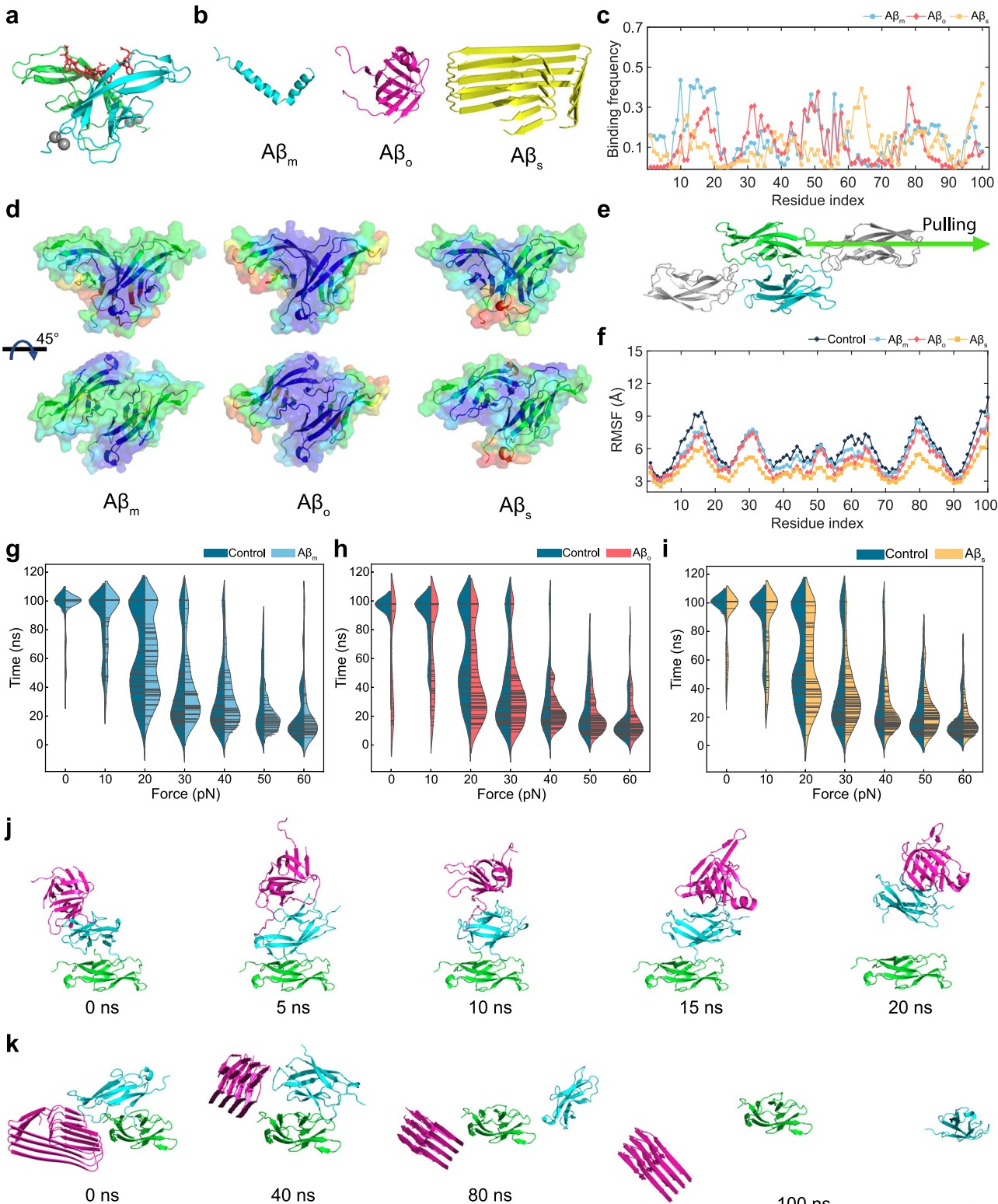

**Fig. 7 | DMD simulations of the binding between Aβ species and an EC1 cadherin dimer, and sDMD simulations of the cadherin-Aβ complexes to characterize Aβ-induced endothelial leakiness. a** Structure of an EC1 cadherin dimer from the full-length of vascular endothelial cadherin (VE-cadherin). Gray spheres and red sticks represent calcium ions and the domain-swapped region, respectively. **b** Structures of a full-length Aβ monomer ($Aβ_m$), oligomer ($Aβ_o$), and seed ($Aβ_s$). **c** Binding frequency of the Aβ species with the EC1 cadherin dimer from the last 10 ns of the binding simulations. **d** Colored binding frequency of the Aβ species with the EC1 cadherin dimer. Blue and red colors on the EC1 dimer surface indicate low to high binding frequencies. **e** Schematic of steered molecular dynamics (sDMD) simulation employed for this study. **f** Root-mean-square fluctuation (RMSF) of the EC1 cadherin dimer with and without the Aβ species. Violin plots for sDMD simulations of the EC1 dimer in the presence and absence of the **g** $Aβ_m$, **h** $Aβ_o$, and **i** $Aβ_s$. Representative dissociation trajectories of the EC1 cadherin dimer with **j** $Aβ_o$ and **k** $Aβ_s$.

formvar/carbon-coated copper grids (400 mesh, Proscitech) and blotted after 1 min incubation. The sample-coated grids were negatively stained with 5 μL of 1% uranyl acetate (UA) for 30 s. The grids were blotted and left to air dry before imaging.

## Attenuated total reflection-Fourier transform infrared (ATR-FTIR) spectroscopy

FTIR spectra (1580–1700 $cm^{-1}$) that contained the amide I regions of each amyloid protein at different stages of fibrillization (monomeric/oligomeric/protofibrillar/fibrillar/seeds) were acquired with an IRTracer-100 (Shimadzu) spectrometer, which was equipped with a He-Ne laser and an MCT detector (Hg-Cd-Te). The MCT detector was constantly being cooled with liquid nitrogen. Regarding sample preparation, the peptide samples were prepared as described above and incubated at 37 °C at 100 μM concentration. The concentration for the fibrillar FapC solution was 50 μM. The collection of the oligomeric/protofibrillar or fibrillar species included a 100 μL acquisition and further lyophilization using a freeze-drier at appropriate time points depending on the aggregation rates of the peptides accordingly ($A\beta_o$: 4 h, $A\beta_{o-p}$: 17 h, $A\beta_f$: 42 h, $\alpha S_o$: 22 h, $\alpha S_f$: 96 h, $IAPP_o$: 1 h, $IAPP_f$: 24 h). Lyophilized samples were further dissolved with 5 μL of MilliQ $H_2O$ and added to the top of the reservoir. Air-drying was then followed using a heat gun. Samples containing monomeric species were in powder form during spectrum acquisition. Data acquisition included the absorbance (%) measurement mode with 512 number of scans. Spectra were acquired with 2–4 $cm^{-1}$ resolution and signals were processed with the Happ-Genzel apodization function. Peak deconvolution that generated the Gaussian band profile for each selected peak on each amide I region was performed with the built-in PeakDeconvolution application through Origin Software (Origin Lab). Deconvoluted band regions were assigned to each type of secondary structure following previous reports[69].

## Cell culture

Human dermal microvascular endothelial cells (HMVECs, catalog number: CC-2543) were obtained from Lonza (Switzerland) and cultured in complete EndoGRO-MV-VEGF growth medium (Merck Millipore, USA). hCMEC/D3 human brain endothelial cells (catalog number: SCC066) were derived from Merck Millipore (USA) and cultured in complete endothelial cell medium (Sciencell, USA). All cell culture was performed under a standard culture condition (37 °C, 5% $CO_2$). In all experiments, endothelial cells were seeded at an initial density of 40,000 cells/$cm^2$ and were cultured to form a confluent monolayer before subsequent treatments.

## Cell viability assay

HMVECs were seeded into 96-well plates and cultured overnight, before being incubated with the amyloid proteins at 20 μM, for durations of 0 h, 0.5 h, 1 h or 2 h. Each group consisted of three biologically independent samples. The cells were subsequently washed with 1× PBS (phosphate-buffered saline). AlamarBlue reagent (Life Technology, USA), prepared in cell media at the recommended concentration by the manufacturer, was added to cells and incubated for 2 h. Fluorescence readings were collected on a microplate reader (Hidex, Finland), at wavelengths of 560/590 nm (excitation/emission). Measurements from all treatment groups were normalized against the measurements from negative (untreated) control group. $H_2O_2$ (200 μM) served as positive control.

## ROS production assay

HMVECs were cultured in 96-well plates and treated with 20 μM of various amyloid proteins for the durations of 0 h, 0.5 h, 1 h, or 2 h. Each treatment group was made up of three independent replicates. After washing with PBS, cells were incubated for 30 min with a mixture of the ROS detector 2′,7′-dichlorodihydrofluorescein diacetate (H₂DCFDA,

1 μM; Merck, USA) and Hoescht 33342 (1 μg/mL; Sigma-Aldrich, USA). Fluorescence measurements were taken on a microplate reader, at excitation/emission of 495/527 nm (H₂DCFDA) and 350/461 nm (Hoescht 33342). H₂DCFDA readings were normalized against respective Hoescht readings to account for cell numbers. Readings from the untreated control group were further used to normalize the other groups' readings. Positive control was $H_2O_2$ (200 μM).

## Immunofluorescence staining of in vitro samples

HMVECs were grown on 8-well chamber slides to reach confluent monolayers. After being treated with different species of amyloid proteins (Aβ 20 μM, IAPP 20 μM, αS 10 μM and FapC 1.6 μM) for 30 min, cells were gently washed with Hank's balanced salt solution (HBSS, Gibco, USA) and fixed by 4% paraformaldehyde (Sigma-Aldrich) for 15 min, followed by permeabilizing and blocking with 0.1% saponin (Sigma-Aldrich) and 5% horse serum (Sigma-Aldrich) in PBS/azide for 1 hour at room temperature. Thereafter, fixed cells were washed three times with PBS and incubated overnight at 4 °C with primary rabbit anti-VE-cadherin antibody (Abcam) at 1:400 dilution with 5% horse serum in PBS/azide. Then, the cells were washed with PBS and incubated with secondary donkey anti-rabbit Alexa Fluor 594 antibody (1:500, Abcam) and Phalloidin-iFluor 488 (1:1000, Abcam) in PBS/azide solution for 2 h at room temperature. After nuclei staining with Hoechst 33342 (Sigma-Aldrich, USA) at 1:2000 dilution for 5 min, the cells were imaged with Leica SP8 lightening confocal microscope (Leica, Germany) through an HC PL APO CS2×63/1.40 oil objective and semiquantitative analysis was performed using ImageJ.

## Treatments prior to exposure to Aβ samples

In assays that involved pre-treatments, the cultured HMVECs were exposed to the respective treatments, which were supplemented into complete EndoGRO-MV-VEGF growth medium, for a duration of 1 h. For experiments with antioxidants, the pre-treatment involved N-acetyl cysteine (NAC, 10 μM; Sigma-Aldrich, USA), which was used to alleviate ROS production. For experiments about endocytosis, the inhibitors monodansyl cadaverine (MDC, 10 μM; Sigma-Aldrich, USA) and methyl β-cyclodextin (MβCD, 5 mM; Sigma-Aldrich, USA) were used as a cocktail to inhibit endocytosis. For experiments related to cell signaling, the Rho-associated protein kinase inhibitor Y27632 (Sigma-Aldrich, USA) and Src family tyrosine kinase inhibitor PP1 (Sigma-Aldrich, USA) were employed at 10 μM in complete growth medium. After the pre-treatment, the medium was replaced with fresh growth medium now containing respective Aβ species and the same antioxidant/inhibitor(s). Growth medium treatment that contained the antioxidant/inhibitor(s) but no Aβ species acted as negative control. The usage of BV-6 antibody as a pre-treatment was described under "competitive inhibition assay" instead.

## Transwell insert assays

In general, HMVECs were cultured on transwell inserts (polycarbonate membrane, 0.4 μm pore diameter; Corning Costar, USA) in a 24-well plate until the formation of a confluent layer (seeded at density of 40,000 cells/$cm^2$, 2 days). HMVECs were treated with different types and concentrations of the amyloid proteins that were supplemented into complete EndoGRO-MV-VEGF growth medium, for the stated durations. Subsequently, Aβ treatments were removed and the wells were washed with PBS. Fresh media with FITC–dextran (1 mg/mL, 40 kDa; Sigma-Aldrich, USA) were then added to quantitatively determine permeability in endothelial barriers. After 30 min exposure to FITC-dextran treatment, solutions in the lower compartment were sampled (100 μL) and their fluorescence was recorded by a microplate reader (Hidex, Finland) at wavelengths of 490/520 nm (excitation/emission). The degree of FITC-dextran transport was defined as fluorescence reading from a treated group normalized by its corresponding untreated control.

## Competitive inhibition assay

Confluent HMVECs cultured on 8-well chamber slides were incubated with different concentrations of VE-cadherin EC1 domain antibody (EC1a, Novus Biologicals, USA), BV-6 antibody (1:20, 1:50, or 1:100; Millipore, USA) in medium containing 3% fetal bovine serum for 1 h. BV-6 antibody specifically recognizes the extracellular domain EC2-3 of VE-cadherin. Cells were washed with HBSS and subsequently incubated with $A\beta_o$ or $A\beta_s$ for 30 min. Furthermore, confluent HMVECs were first incubated with $A\beta_o$ or $A\beta_s$ for 30 min, then EC1a or BV-6 antibody at different concentrations was added and incubated for 1 h after washing the cells with PBS. Then, immunofluorescence staining was performed to observe the cell leakiness. IgG (Sigma-Aldrich, USA) of 1 μg/mL with 1 h incubation was used as a negative control.

## Association of Aβs and VE-cadherin assay

Confluent HMVECs were incubated with ThT-labeled $A\beta_s$ for 30 min, which were derived from the fibrils in the ThT assay. After that, excess $A\beta_s$ were removed, and the cells were washed with HBSS. Immunofluorescence staining was performed to observe the location of ThT-labeled $A\beta_s$ and VE-cadherin.

## Immunoblotting

HMVECs were cultured in 6 cm cell culture dishes and subjected to the pre-treatment of PP1 inhibitor in fresh cell medium or fresh medium only. Subsequently, HMVECs were exposed to different concentrations of $A\beta_o$ and $A\beta_s$ for 1 h. Following the exposure, each dish was washed thrice with chilled PBS and lysed with Laemmli sample buffer (63 mM Tris-HCl pH 6.8, 2% sodium dodecyl sulfate (SDS), 10% glycerol, 1% 2-mercaptoethanol, and 0.0005% bromophenol blue). The cell lysis buffer solution had been supplemented with a cocktail of 1% protease and phosphatase inhibitors (Sigma-Aldrich, USA). Gel electrophoresis was conducted with the derived cell lysates using 10% resolving polyacrylamide gels (Mini Protean, Bio-Rad, USA) and proteins transferred onto nitrocellulose membranes (Sigma-Aldrich, USA). The membranes were blocked with 5% bovine serum albumin (BSA, Sigma-Aldrich, USA) solution for 1 h and incubated with a solution of the relevant primary antibody overnight at 4 °C. Subsequently, membranes were washed thrice and then incubated in a solution of relevant horseradish peroxidase (HRP)-conjugated secondary antibody for 1 h. Membranes were exposed to Immobilon Western Chemiluminescent HRP substrate kit (Merck, USA), and protein bands were captured visually through a chemiluminescence imaging system (Syngene, UK). Expression levels of proteins in images were analyzed semi-quantitatively through ImageJ software and normalized against the respective control group's protein band within each image. In all washing steps, as well as preparation of blocking and antibody solutions, Tween 20 detergent (TBST; composed of 150 mM NaCl, 20 mM Tris-HCl, 0.1% Tween 20) was used. Primary antibodies were used at dilution of 1:1000 and secondary HRP-conjugated antibodies were employed at 1:2500. The complete list of antibodies utilized for immunoblotting is provided in Supplementary Table 2.

## Immunoprecipitation assay

Confluent layers of HMVECs were treated according to specific experiment designs. At the conclusion of treatment, the cells were washed with PBS thrice and lysed with RIPA extraction buffer (Thermo Fisher, USA) containing a cocktail of protease/phosphatase inhibitors. Samples were subsequently handled in the immunoprecipitation assay buffer (20 mM HEPES buffer pH 7.5, 150 mM NaCl, 0.1% Triton X-100, and 10% glycerol), where relevant immunoprecipant antibody was incubated with the sample for 90 min at suitable dilution, with gentle mixing. The protein A/G PLUS–agarose beads (Santa Cruz Biotechnology) were subsequently added. Samples were centrifuged and washed with immunoprecipitation assay buffer for a total of four times. The final spun-down pellet was resuspended in SDS loading buffer and proceeded to be subjected to immunoblotting/western blot assays according to the method described above.

## In vivo leakiness of APP/PS1 mice

All in vivo mice experiments were approved by the Southwest University Animal Care and Use Committee. Experiments were carried out according to the NIH's guidelines for the care and use of laboratory animals. The mice were supplied with free access to food and water and were kept at $22 \pm 2$ °C with $50 \pm 10\%$ humidity environment, and light/dark cycle of 12 h. APP/PS1 and C57BL/6 J mice, 2 months and 12 months old, were obtained from Ziyuan Laboratory Animal Technology Co., LTD (Hangzhou, China). An equal number of male and female mice per group was used for this assay. To assess the integrity of the vessels in APP/PS1 mice, the mice received an intravenous injection of 10 mM EBD solution (MCE, China) of 100 μL. After 24 h, the mice were sacrificed to obtain the tissues for imaging by using NEWTON 7.0 Imaging System. The mice without EBD injection were used to blank the basal fluorescence of the tissue itself during the experiments.

For the quantitative in vitro determination of Aβ concentrations in plasma and tissues of APP/PS1 and C57BL/6 J mice, a mouse $A\beta_{42}$ ELISA kit (Mlbio, China) was used in the assay. Blood samples were collected from the orbital venous plexus, and heparin was used as an anticoagulant. The samples were centrifuged for 15 min at $1000 \times g$ at 4 °C. Plasma was collected and stored in aliquot at $-80$ °C for later use. Then, the mice were sacrificed, and their tissues were collected. After that, the tissues were washed with ice-cold PBS to thoroughly remove excess blood. Then the tissues were homogenized in PBS on ice, followed by sonication using an ultrasonic cell disrupter to further break the cells. The homogenates were then centrifugated for 5 min at $5000 \times g$ to obtain the supernatants. The standard curve and tissue samples were determined according to the protocol of the kit, and the absorbance was measured at 450 nm.

## In vivo leakiness assay of Aβs and Aβo

Adult, 10 weeks old, Swiss mice were obtained from Leask Laboratory Animals Co, Ltd. (Shanghai, China). An equal number of male and female mice per group was used for this assay.

a. Mice received once intravenous injections of 0.002, 0.02, 0.2, 2, 20, 50 or 100 μg $A\beta_s$- or $A\beta_o$-containing EBD solution (10 mM, 100 μL). The control mice received an intravenous injection of 10 mM EBD solution. An equal number of 3 male and 3 female mice per group was used for this assay.

b. Mice received intravenous injections of 2 or 20 μg $A\beta_s$, $A\beta_o$ or $IAPP_s$ of 100 μL. As another permeabilization indicator, fluorescein isothiocyanate-dextran (FITC-dextran, 10 mg/mL, 10,000 Da, MCE, China) was injected 30 min before sacrificing the mice.

c. The assay concluded four groups, PP1, $A\beta_s$, PP1 (1 h) + $A\beta_s$, and $A\beta_s$ (3 h) + PP1. PP1 (1 h) + $A\beta_s$ refers to pre-injection of PP1 at 1 h prior to $A\beta_s$ administration, and $A\beta_s$ (3 h) + PP1 refers to 3 h administration of $A\beta_s$ followed by PP1 injection. The mice received an intravenous injection of 2 or 20 μg $A\beta_s$-containing 10 mM EBD solution of 100 μL. Src-kinase inhibitor, PP1, was injected through the tail vein at 1.5 mg/kg/mouse of 100 μL. The EBD fluorescence intensity of the brain organs was measured at 24 h-administration of $A\beta_s$.

The mice without EBD or FITC-dextran injection were used to blank the basal fluorescence of the tissue itself during the experiments. For the control groups, the mice received the same amount of EBD or FITC-dextran as the sample groups. After 24 h of amyloid species injection, the mice were sacrificed to obtain tissues, including tissues from the brain, the heart, the liver, the lungs, the spleen, the kidneys, and the diaphragm, for imaging by using NEWTON 7.0 or IVIS® Lumina III Imaging System. Mice tissue fluorescence data were analyzed using EvolutionCapt-v18.02 and Living Image® 4.3.1 software. The mean signal intensity (photons/second/cm²/steradian) of EBD or FITC-dextran was acquired. The fluorescence intensities of the sample

groups were normalized to the corresponding mean values from the control group.

## Immunofluorescence staining of mice brain samples

After the imaging of in vivo leakiness, the brains were fixed in 4% paraformaldehyde solution and removed after 48 h. Sucrose (30%) was added overnight for dehydration after brain fixation. They were then embedded with optimal cutting temperature (OCT) compound and sliced in 5 µm-thick transverse sections by a freezing microtome. Then, slices were incubated with 10% skimmed milk for 2 h, then incubated with anti-Aβ antibody (1:250 dilution) overnight. Alexa Fluor 488 or 647-conjugated secondary antibody (1:250 dilution) was used for fluorescence detection. DAPI was used for the visualization of nuclei and EBD was also observed at 555 nm with a super-resolution laser confocal microscope (Nikon, N-SIME).

## DMD simulations for amyloid and EC1 cadherin dimer binding

All-atom discrete molecular dynamics (DMD) simulation with implicit solvent models was used for this study to characterize the Aβ induced VE-cadherin dissociation. DMD is a unique category of molecular dynamics (MD) with significantly enhanced sampling efficiency, which has been widely applied to biomolecular studies such as protein folding, peptide aggregation[70–72], and understanding the protein structure and dynamics[73,74]. Inter-atomic potential for DMD simulation consisted of the bonded (i.e., bonds, bond angle, and dihedral angle) and non-bonded terms (hydrogen bonds, solvation, electrostatic, and van der Waals). For the non-bonded terms, hydrogen bond was remodeled with reaction-like algorithm[75] and EEF1 implicit model by Lazaridis and Karplus was used for solvation[76]. Debye-Hückel approximation and CHARMM forcefield[77] were applied to van der Waals and electrostatic terms. From our recent study, it has been demonstrated that the EC1 dimer from full-length VE-cadherin is a suitable model to characterize the *trans* interaction mimicking the cadherin dimer coming from two opposing cells. Therefore, the EC1 cadherin dimer adapted from the cryo-EM model of EC1-2 cadherin dimer was considered (PDB ID: 3PPE[78]) for the current study. For constructing the molecular model of EC1 dimer for this study, the bond constraints for calcium ion sites (i.e., residues Glu11, Asp62, Glu64, Asp96, and Asp99) and Gō-potential were applied to the domain-swapped region of the EC1 dimer, respectively (Fig. 7a). Specifically, the weak contact energy of 0.4 kcal/mol (-0.67 $K_BT$) was assigned to $C_\beta$ atoms of contacting residues. Next, we prepared three different forms of $A\beta_{42}$ monomer ($A\beta_m$), oligomer ($A\beta_o$), and tetramer seed ($A\beta_s$). The atomic models of $A\beta_m$ (PDB ID: 1IYT[79]) and $A\beta_s$ (PDB ID: 5OQV[80]) were used, and $A\beta_o$ was brought from our recent work[81]. To consider the $A\beta_s$ model, the seed was relaxed and equilibrated for 50 ns with the application of Gō-constraints between each peptide. The same contact energy of Gō-potential applied on EC1 dimer was assigned to the inter-peptides of $A\beta_s$. Each amyloid peptide was randomly located away from the EC1 dimer, at least 12 Å away in a 150 $nm^3$ cubic box, and counter ions were distributed to neutralize the net charge. To avoid a biased potential energy, different initial velocity was applied, and 40 independent DMD simulations each of 50 ns (an accumulative 2.0 µs DMD simulations) were performed. 50 fs/step of the unit simulation time and 1 kcal/mol of corresponding energy were employed and a temperature of 300 K was maintained with Anderson's thermostat. After the binding simulations, we computed the binding frequencies of the amyloid proteins with the EC1 dimer from the last 20 ns of binding simulations. To calculate the binding frequency, we assigned 0.65 nm of cutoff distance to get an atomistic contact between the EC1 cadherin dimer and amyloid proteins.

## Steered discrete molecular dynamics (sDMD) simulations

We employed sDMD simulations to identify the effects of amyloid proteins on VE-cadherin dimer disruption. This constant force-pulling in silico experiment generally mimics force spectroscopy methods such as atomic force microscopy and optical tweezers. With respect to applying either a constant force or a velocity, this technique enables the characterizations of protein unfolding, protein structure, and dynamics. To carry out the sDMD simulation, we immobilized one of the EC1 domains and made flexible of the other domain[57]. Constant forces were applied to the flexible domain of the EC1 cadherin dimer towards the EC2 cadherin dimer (Fig. 7e). 10 pN of interval forces in the range of 0 ~60 pN was given during the sDMD simulations. For sufficient sampling, 70 cases of independent sDMD simulations each for 100 ns were performed. The detailed conditions for running the sDMD simulations were the same as for the binding simulation of the cadherin dimer and amyloid proteins. The sDMD simulation details are also listed in Supplementary Table 3.

## Statistics and reproducibility

The in vitro assays, including ROS production, in vitro transwell assays, immunoprecipitation, western blotting, and confocal fluorescence microscopy, were derived from at least three biological samples. The cells or conditions were assigned randomly to each experimental group. The extent of endothelial leakiness was expressed by gap area and distribution, which were derived from the images using the trainable Weka segmentation plugin in ImageJ 1.53c software. The in vivo experiments were performed with at least three animals for each sample condition. Mice were randomly grouped before different treatments.

Data graphing and statistical analysis were performed with GraphPad Prism version 9.3.1 (GraphPad Software, La Jolla). All data were expressed as mean ± standard deviations (SD), analyzed via two-tailed Student's *t* test, one-way or two-way ANOVA, and followed by Tukey's multiple-comparisons test as indicated in the respective figure captions. $P < 0.05$ was considered statistically significant.

## Reporting summary

Further information on research design is available in the Nature Portfolio Reporting Summary linked to this article.

## Data availability

The source data underlying Figs. 1–6 and Supplementary Figs. 14, 6, 8–14, 16–20, 25, 26 are provided in the Source Data file. The simulation data for Fig. 7 and Supplementary Figs. 21–24 are deposited to Zenodo at: https://doi.org/10.5281/zenodo.10152860.

The atomic models of EC1 cadherin dimer, $A\beta_m$, $A\beta_s$ are available from the database PDB ID: "3PPE", "1IYT", and "5OQV", respectively. Source data are provided with this paper.

## Code availability

The discrete molecular dynamics (DMD) simulation code and analysis script used to generate the results reported in this manuscript are available at Molecular in Action, LLC, moleculesinaction.com/home.html and https://doi.org/10.5281/zenodo.10152860.

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

## Acknowledgements

This work was supported by the National Natural Science Foundation of China (T2250710182 P.C.K., 82104087 Y.L., 21976145 Y.S., and 21974110 Y.S.), National Key Research and Development Program, Ministry of Science and Technology of China (2022YFC2409700 PCK, 2021YFA12009000 P.C.K.), National Institutes of Health (R35GM145409 F.D. and P20GM121342 F.D.), and the Singapore Food Agency Singapore Food Story Program (W22W3D0007 D.T.L.). The authors thank Scott Peng and Kairi Koppel for their technical assistance.

## Author contributions

P.C.K. and D.T.L. conceived the project. P.C.K., D.T.L., Y.L., T.P.D., Y.S., and F.D. designed the project. Y.L., N.N., M.L., and P.C.K. wrote the manuscript. Y.L., N.A., and A.K. performed TEM, FTIR, confocal fluorescence microscopy, competitive inhibition, in vitro transwell assays, as well as gap area analysis. N.N. performed in vitro transwell, signaling pathway, viability, ROS, endocytosis, and immunoprecipitation assays. Y.L. and W.W. performed in vivo assays. M.L. performed DMD and sDMD simulations. All authors discussed and agreed on the presentation of the manuscript.

## Competing interests

The authors declare no competing interest.
