## [Peer Review File · Nature Communications]

Endothelial Leakiness Elicited by Amyloid Protein AggregationREVIEWER COMMENTS

Reviewer #1 (Remarks to the Author):

In this work by Li Y and Ni N et. al., authors address the possibility of seeds of pathogenic amyloid peptides to act as inducers of endothelial leakiness (EL). They show that purified amyloid peptides can aggregate and can give rise to multiple forms of oligomer/fibrillar structures in vitro. Interestingly, anionic amyloid oligomers were capable of inducing EL in short time periods, a response independent of endocytosis/ROS/apoptotic signaling.

Authors propose that the EL response is due to direct interaction of A β 42 with VE-cadherin on endothelial surface and potentially involves the EC1 dimer interface of VE-cadherin with that of A β 42 oligomers using simulation studies. Finally, authors claim that injection of A β 42 alone is sufficient to induce EL within the brain of mice in vivo. Overall, the study provides a new dimension to the pathogenic potential of amyloid peptides, specifically A β 42 in the context of AD, where interaction of such amyloid aggregates with endothelial cells might independently induce EL in vivo.

While the conceptualization, experimental design and analysis presented are impressive, a few shortcomings need to be addressed (see below) to make authors' conclusions justified and standout.

Major concerns

1) Why do A β 0-p2 and A β f fail to elicit EL, is it due to their inability to interact with VE-cadherin? It is important to establish why fibrillar form might be less potent than oligomeric/sonicated form.

2) Clone BV-6 blocks VE-cadherin function via its mapped interaction with EC2-3 portions of the extracellular portion, whereas authors describe simulated effects of A β s on EC1 dimers. Unless A β 42 potentially interacts with all three domains on the extracellular portion of cadherin, it is difficult to reconcile with the reported observations. To be more consistent with their simulations, authors could alternately use Clone Cad5, which was mapped to EC1 domain of cadherin extracellular domain.

3) Authors also need appropriate controls for experiments with clone BV-6 to establish that

the blocking effects might not be observed using say an IgG control or an antibody against RAGE (another possible membrane receptor for A β 42). Antibody against RAGE will be important to establish the specificity of A β 42 for VE-cadherin.

4) In their pulldown assays, authors need to show that A β 42 specifically interacts with VE-cadherin and not other potential surface proteins including RAGE as well as tight junction proteins such as Occludin or Claudin. Same applies to their confocal experiments showing overlap between VE-cadherin and A β 42.

5) In their studies with APP/PS1 mice, authors must include a background strain control for the measurement of A β 42 amounts in young and old mice. This is required to distinguish aging-related deposition from that of AD-specific disease progression phenotypes.

6) The use of EBD to study blood-brain barrier functions has been discouraged widely. In this study, authors measure EBD signals 24 h post its injection. I suggest authors to use fluorescent dextran or quantum dots to validate their results observed from EBD injection, especially to the points mentioned below –

a) A β 42 induces EL phenomena in vitro in a short time (~30 mins). If indeed A β 42 aggregates act quick and is independent of other inflammatory signals, one would expect EL to occur earlier in vivo, say 2-3 h and not 24 h post A β 42 administration. Can the authors check for EL at 3 h and 24 h in the brain by injecting fluorescent dextran/quantum dots 30 mins – 1h before measurement.

b) To establish specificity of effects seen with A β 42, authors should use IAPP (2 ug) and vehicle control, and measure EL as suggested above in (a) at 3 & 24 h.

7) Lastly, A β 42 appears to induce EL specifically in the brain and diaphragm. However, the in vitro EL effects appear general/non-discriminatory. Does A β 42 have additional preference for EL in vivo? Do authors know if A β 42 they inject gets deposited in the brain and diaphragm, as against other organs they have tested for?

Minor concerns

1) What is the justification for using only male mice for experimentation using the A β 42 injection and subsequent measurements involving EBD?

2) The authors report in vivo results as n=3 independent mice, are these from a single experiment? If so, they need to biologically replicate their experiments for reproducible results and reliable statistical measures.

Reviewer #2 (Remarks to the Author):

Amyloid formation is a key phenomenon underlying neurodegenerative diseases such as Alzheimer's disease (AD). Understanding the pathogenic mechanisms of the amyloid formation process is important for illustrating the complex causes of AD and could aid the development of new therapeutic strategies.

Li et al. found that amyloid protein oligomers and fibril seeds can lead to endothelial leakiness (APEL), analogous to nanomaterials-induced endothelial leakiness (NanoEL). The size and anionic charge properties of A β aggregates are important for APEL. This process involves extracellular interactions between A β aggregates and vascular endothelial (VE)-cadherin, which triggers a cascade of signaling events and actin remodeling, leading to endothelial leakiness. The authors validated APEL in ex vivo blood vessels and in vivo blood-brain barrier models. Using discrete molecular dynamics simulation tools, the authors characterized some potential binding and functional mechanisms between A β aggregates and VE-cadherin. This work is interesting and has some novel insights into the role of A β aggregates in AD-related vascular permeation. However, there are some concerns that the authors need to address:

Major concerns:

1) In Figure 2, the authors showed that A β oligomers, protofibrils, and seeds could lead to endothelial leakiness, while the monomer and fibril forms of A β cannot. The authors expanded this conclusion to other anionic amyloid proteins like alpha-synuclein, islet

amyloid polypeptide, and FapC. However, though the fibril forms of these three proteins were prepared in Figure 1, they were not tested in Figure 2. Without showing the result of their fibril forms, i.e., α Sf, IAPPf, and FapCf, how can the author conclude 'their soluble monomers and lengthy amyloid fibrils were unable to induce leakiness'?

2) There have been many studies regarding the endothelial toxic effects of A β aggregates. The authors found that the APEL process is unique because it does not increase ROS and apoptosis. The reason could be short-term (30 mins) treatment. The result is very interesting. However, the impact of A β aggregates is usually long-term in physiological conditions. It could be possible that APEL is an earlier event before the long-term toxic effect. However, most (if not all) of the cell-based experiments in the study are a short-term treatment of A β aggregates (30mins to 2 hours) and do not extend to long-term exposure, for example, 24-72 hours that other researchers typically use. Therefore, it is hard to discuss the relationship between the short-term and long-term effects. For example, does the long-term effect found in previous reports also involve VE-cadherin signaling events? Can correcting APEL at the early stage reduce long-term toxicity, such as ROS production or apoptosis? A better understanding of the relationship between APEL and the previously reported model(s) could strengthen the novelty of APEL.

3) In lines 321-322, the author stated, 'Although adding BV-6 may also lead to some significant gap formation in HMVEC monolayers, it only occurred at a much higher BV-6 concentration (dilution of 1:20)'. However, in Figure 4B, the gap area of BV-6 treatment at 1:100 concentration is also significantly higher than media control. Please explain this result that conflicts with the above description. Furthermore, as BV-6 and A β o/A β s all lead to a higher gap area compared to media control, could it be possible that the decrease of gap area observed in (BV-6 1:100 + A β o) condition is due to the possible interactions between BV-6 and A β aggregates rather than through the VE-cadherin pathway?

4) For the immunoprecipitation assay described in lines 338-342, the authors demonstrated the physical binding between A β o and VE-cadherin. Did the authors compare the binding of VE-cadherin to A β m, A β s, and A β f using the immunoprecipitation assay? The comparison would be necessary to test whether VE-cadherin is essential for different APEL phenotypes

triggered by different A β aggregation forms. Furthermore, the results may help to support/compare the in-silico modeling observations.

5) The authors used APP/PS1 mice or Swiss mice injected with A β s or A β o to validate the endothelial leakiness in vivo. However, whether or not the observed leakiness is through the VE-cadherin pathway has not been tested. BV-6 antibody or Src-kinase inhibitor, PP1, can be used to check whether the endothelial leakiness phenomenon can be reversed in vivo. These experiments can indicate more clearly the novelty that sets this work different from other studies using evidence from both in vitro and in vivo. These experiments can also highlight the therapeutic potential of their discovery.

6) For the discrete molecular dynamic simulation, the structure of one molecule of A β m, A β o, or A β s was used for modeling the interaction with an EC1 cadherin dimer. How do the authors know the binding ratio between A β species and EC1 cadherin dimer is 1:1? Would the results be different if using two or more molecules of A β species for the modeling?

7) Does the simulation study explain the requirement of anionic charges for amyloid protein to trigger endothelial leakiness? A β m, A β o, and A β s all show different binding mechanisms in the simulation results. A β o shows strong binding properties to the cationic turn regions of the EC1 dimer, which may explain the need for anionic charge in A β species. However, A β s that also triggers endothelial leakiness shows a different binding pattern. The authors illustrated the different binding frequencies around the EC1 dimer surface (Fig.7D) but did not describe the binding properties on the structure of A β species to explain the earlier experimental results.

8) In the simulation study, the author found that A β o significantly increases cadherin dimer's dissociation, while A β m has a negligible impact on the dimer dissociation. The author suggests that the observation agreed well with the experimental data. However, is cadherin dimer's dissociation or stability involved in the APEL process? There is no direct experimental evidence in the manuscript supporting this.

Minor points:

1) Line 195, 'protofirbils' should be 'protofibrils'.

2) Line 300, the figure legend indicates 'green: actin', but there is no green in Figure 4A.

3) It seems that it is better to describe result section 2.7 before result section 2.6 if some of the in-silico modeling results can be validated using ex vivo or in vivo models.

Reviewer #3 (Remarks to the Author):

Li and colleagues suggest an attractive hypothesis regarding the roles of amyloid- β ($A\beta$) aggregates in the pathogenesis of Alzheimer's disease (AD) that extracellular interactions between $A\beta$ aggregates and VE-cadherin are responsible for the endothelial leakiness, eventually causing pathogenesis of AD. The authors conducted this research not only on experimental methodology, including in vitro, ex vivo, and in vivo assays, but also in silico methodology to clarify the mechanism of endothelial leakiness caused by aggregates of $A\beta$ and several amyloidogenic proteins. I firmly believe that this manuscript attracts broad-spectrum readers because of its importance in the AD pathogenesis. However, several concerns have arisen after my review. The authors should address five points as follows.

1. Throughout the manuscript, the authors observed endothelial leakiness with aggregates formed using protein solutions with high concentrations, mainly 20-40 μ M, compared with its biological concentrations. Can authors rationalize that the mechanism regarding endothelial leakiness in assays performed in vitro can be the same as in vivo ones? The authors should clarify the difference in the concentrations of protein between in-vitro assays and in-vivo environments and discuss the possible discrepancies in the mechanisms of endothelial leakiness, if any.

2. Throughout the manuscript, although the authors insist that interactions between $A\beta$ aggregates and VE-cadherin are important in the mechanism on which protein aggregates cause endothelial leakiness, the data regarding the dissociation constant of the interaction are not shown in the manuscript. Do the interactions happen in a specific or non-specific manner? I understand that the authors tried to reveal this point using a BV-6 antibody (Figure 4B). But adding the BV-6 antibody itself induced endothelial leakiness (the orange bar in Figure 4B), raising the question about the specificity of APEL. Authors should clarify the dissociation constant of the interaction, preferably with experimental data of interaction analyses. Otherwise, readers cannot understand the specificity of the endothelial leakiness caused by protein aggregates.

3. In Figure 1B, the authors defined several species of aggregates on their aggregation pathways. But the definition is ambiguous. For example, what is the difference between $A\beta_o$, $A\beta_{o-p1}$, and $A\beta_{o-p2}$? I recommend the authors clarify the clear differences between identical species.

4. In Figure 1G, the authors prepared four kinds of species, monomer (m), oligomer (o), fibrils (f), and seeds (s), for three kinds of proteins, $A\beta$, α S, and IAPP. However, the data

showing a gap area formed by identical aggregates (Fig. 2A, B, and C) do not include fibril species regarding α S and IAPP. The authors should show the data of these two species (α S_f and IAPP_f) to generalize the hypothesis that the breakdown of mature fibrils into shorter seeds is important to induce endothelial leakiness.

5. In section 2.6 (Fig. 6), the authors suddenly increased the concentration of A β species. The authors should clarify the reason why they increased the concentrations of proteins. Furthermore, the morphologies of aggregates highly depend on solution conditions (i.e., initial concentrations of proteins). I am afraid that A β ₁₋₄₂ peptide would form amorphous globular aggregates with concentrations on the order of 100 μ M. Thus, the authors should confirm the characteristics of aggregates formed in higher concentrations of protein monomers.

Response Letter

We thank the reviewers for their unanimous endorsements of the significance and novelty of our manuscript. We also wish to thank the reviewers for their insightful constructive comments, which have been fully incorporated in the revised manuscript and have strengthened the rigor and presentation of our study. All changes are highlighted in red in the revised main text and the SI for easy viewing.

In this revision, to fully address the comments made by the reviewers, we have:

- 1) added IAPP_f and α S_f as control for APEL (Fig. 2);
- 2) performed EC1a blocking of VE-cadherin *in vitro* (Fig. 4A-C);
- 3) performed an additional co-immunoprecipitation assay for probing the interactions between VE-cadherin and the four forms of A β following HMVECs exposure to A β _o (Fig. 4E);
- 4) examined the reversibility of APEL *in vivo* (Fig. 6F&G);
- 5) performed a transwell assay for the protein aggregate species we did not examine in the original submission (Fig. S2);
- 6) performed blocking with EC1a *in vitro*, in the opposite incubation sequence to that in Figure 4A-C (Fig. S7);
- 7) added IgG control (Fig. S8);
- 8) examined the interactions of tight junction proteins with A β _o (Fig. S10);
- 9) performed detection of A β ₄₂ levels in the tissues and blood of C57BL/6J mice (Fig. S12 C&D);
- 10) performed an *in vivo* experiment of A β _s/A β _o-induced endothelial leakiness with an equal number of male and female mice (Fig. S14);
- 11) examined the timeframe of A β _o/A β _s-induced vascular leakiness *in vivo* with FITC-dextran as an alternative permeabilization indicator (Fig. S18);
- 12) performed detection of IAPP-incited vascular leakiness *in vivo* (Fig. S19);
- 13) added details of steered discrete molecular dynamics (SDMD) simulation conditions (Table S3)
- 14) provided dissociation rates of cadherin dimer under different forces in both presence and absence of A β aggregation species derived from SDMD simulation (Table S4).

Reviewer #1 (Remarks to the Author):

In this work by Li Y and Ni N et. al., authors address the possibility of seeds of pathogenic amyloid peptides to act as inducers of endothelial leakiness (EL). They show that purified amyloid peptides can aggregate and can give rise to multiple forms of oligomer/fibrillar structures *in vitro*. Interestingly, anionic amyloid oligomers were capable of inducing EL in short time periods, a response independent of endocytosis/ROS/apoptotic signaling. Authors propose that the EL response is due to direct interaction of A β ₄₂ with VE-cadherin on endothelial surface and potentially involves the EC1 dimer interface of VE-cadherin with that of A β ₄₂ oligomers using simulation studies. Finally, authors claim that injection of A β ₄₂ alone is sufficient to induce EL within the brain of mice *in vivo*. Overall, the study provides a new dimension to the pathogenic potential of amyloid peptides, specifically A β ₄₂ in the context of AD, where interaction of such amyloid aggregates with endothelial cells might independently induce EL *in vivo*. While the conceptualization, experimental design and analysis presented

are impressive, a few shortcomings need to be addressed (see below) to make authors' conclusions justified and stand out.

We thank the reviewer for this summary and strong endorsement of our work.

Major concerns

1) Why do A β _{0-p2} and A β _f fail to elicit EL, is it due to their inability to interact with VE-cadherin? It is important to establish why fibrillar form might be less potent than oligomeric/sonicated form.

Thank you for this interesting question. Endothelial leakiness in HMVEC monolayers was observed upon their exposure to the nanoparticulates of oligomers, protofibrils, and sonicated amyloid seeds of A β , mediated by their interactions with adherens junction as revealed by our study. The physical gap between apposing endothelial cells is approx. 20 nm (Dejana et al. *J. Cell Sci.* 121, 2115, 2008), while the elasticity and fluidity of cell membranes affords more tolerance of larger nanoparticulates up to ~100 nm per literature (Tee et al. *Chem. Soc. Rev.* 48, 5381, 2019). In comparison with the oligomers, seeds, and early-stage protofibrils that can be accommodated by such small paracellular space, amyloid fibrils are sub-microns to tens of microns in length and are too rigid (Ke et al. *Chem. Soc. Rev.* 49, 5473, 2020) to fit in the paracellular endothelial space between to elicit APEL.

2) Clone BV-6 blocks VE-cadherin function via its mapped interaction with EC2-3 portions of the extracellular portion, whereas authors describe simulated effects of A β s on EC1 dimers. Unless A β ₄₂ potentially interacts with all three domains on the extracellular portion of cadherin, it is difficult to reconcile with the reported observations. To be more consistent with their simulations, authors could alternately use Clone Cad5, which was mapped to EC1 domain of cadherin extracellular domain.

Thank you for this question. According to this comment we have further employed a VE-cadherin antibody, referred to as EC1a, developed using only the EC1 domain of VE-cadherin. The EC1a was developed against recombinant protein corresponding to the sequence: SLPHHVGKIKSSVSRKNAKYLLKGEYVGKVFVRVDAETGDVFAIERLDRENISEYHLTA VIVDKDTGENLETP. The competitive inhibition assay set up between EC1a and A β species revealed that the percentage of gap formation was significantly reduced at the EC1a dilutions of 1:100 and 1:500 pre-treatment (EC1a+A β _o/A β _s), compared to A β _o/A β _s only (**Figure 4**). As the EC1a antibody concentration was increased, the VE-cadherin junction between adjacent cells became unstable as the antibody EC1a itself destabilized the cell junction and gradually broke apart to form gaps, especially at the highest EC1a concentration (1:20 dilution). Similar phenomena were observed with the incubation of A β _o/A β _s and BV-6 antibody, indicating that both the extracellular EC1 and EC2-3 domains of VE-cadherin were important for the occurrence of APEL.

Our binding simulation of EC1-2 dimer with different A β species (**Figure S23**) revealed that A β aggregates could indeed bind different EC domains. Hence, the simulation results are consistent with the APEL block assays with both EC1a and BV-6 (**Figures 4&S8**). Since the binding-induced disruption of EC dimer was mainly driven by the allosteric effect that

propagated to the dimer interface at the EC1 domain as shown by our prior study (Lee *et al. Adv. Sci.* **8**, e2102519, 2021), we focused on the first domain in our steered DMD simulations to characterize the binding induced disruption of the VE-cadherin dimer *in silico*.

3) Authors also need appropriate controls for experiments with clone BV-6 to establish that the blocking effects might not be observed using say an IgG control or an antibody against RAGE (another possible membrane receptor for A β ₄₂). Antibody against RAGE will be important to establish the specificity of A β ₄₂ for VE-cadherin.

Thank you for this comment. We have incorporated an IgG control in the measurement of the blocking effect. There was no APEL occurrence under the treatment of IgG with A β ₀ at 1 μ g/mL for 1 h (**Figure S8A**). Anti-RAGE antibody was also used in the immunofluorescence assay, and there was no RAGE expression observed in HMVECs (**Figure S10**).

4) In their pulldown assays, authors need to show that A β ₄₂ specifically interacts with VE-cadherin and not other potential surface proteins including RAGE as well as tight junction proteins such as Occludin or Claudin. Same applies to their confocal experiments showing overlap between VE-cadherin and A β ₄₂.

Thank you for your suggestion. Accordingly, we have included additional co-immunoprecipitation assays to probe A β ₀-treated endothelial samples for interactions that might involve proteins claudin-5 and occludin (**Figure S10**) with A β ₀. It was observed that there were no obvious interactions between A β ₀ and claudin-5 or occludin under our APEL experiment conditions and time frame.

Secondly, we noted that RAGE was not detected at observable levels for the cell type (i.e., HMVECs) in our western blot assays. Our data appeared to be in general agreement with the literature where high RAGE expression levels are found in AD patients' neuronal membrane. It appears that RAGE is primarily expressed on endothelial cells during other non-AD diseases such as diabetes (Xiong *et al. J. Biol. Chem.* **286**, 35061, 2011; Wang *et al. Sensors* **17**, 722, 2017; Kierdorf *et al. J. Leuko. Biol.* **94**, 55, 2013; Kay *et al. J. Diabetes. Res.* e6809703, 2016). In light of the minimum RAGE detected and unlikeliness of RAGE being drastically upregulated within the ~30 min time frame of APEL, interactions at RAGE were unlikely to be a significant contributor to APEL.

5) In their studies with APP/PS1 mice, authors must include a background strain control for the measurement of A β ₄₂ amounts in young and old mice. This is required to distinguish aging-related deposition from that of AD-specific disease progression phenotypes.

Thank you for this comment. We have incorporated background strain mice (C57BL/6J, 2 months and 12 months) of the same ages as our AD mice as control for the measurement of A β ₄₂ amounts (**Figure S12**).

6) The use of EBD to study blood-brain barrier functions has been discouraged widely. In this study, authors measure EBD signals 24 h post its injection. I suggest authors to use fluorescent dextran or quantum dots to validate their results observed from EBD injection, especially to the points mentioned below –

Thank you for the comment. We have performed the suggested fluorescent dextran experiments as detailed below.

a) A β 42 induces EL phenomena *in vitro* in a short time (~30 mins). If indeed A β 42 aggregates act quick and is independent of other inflammatory signals, one would expect EL to occur earlier *in vivo*, say 2-3 h and not 24 h post A β 42 administration. Can the authors check for EL at 3 h and 24 h in the brain by injecting fluorescent dextran/quantum dots 30 mins – 1h before measurement.

According to the suggestion, we have tested EL *in vivo* at 3 h and 24 h in the brain (**Figure S18**). FITC-dextran (10,000 Da) was injected 30 min before the measurement. We did not observe significant APEL *in vivo* at 3 h administration with 2 and 20 μ g of A β_o /A β_s . With time extended to 24 h, APEL did occur in the brain under the treatment of A β_o /A β_s . Indeed, while A β amyloid toxicity may develop in ~24 h *in vitro* it takes months to materialize in mice and years or decades in human due to the complex anatomies and physiologies *in vivo*. The 24 h timepoint chosen for our *in vivo* observation was, therefore, not unreasonably long as the extrapolation of observations from *in vitro* to *in vivo* is typically nonlinear in timescale.

b) To establish specificity of effects seen with A β 42, authors should use IAPP (2 μ g) and vehicle control, and measure EL as suggested above in (a) at 3 & 24 h.

Thanks for the suggestion. We have incorporated 2 and 20 μ g of IAPP for *in vivo* experiments (**Figure S19**). The vehicle control in this study was PBS (1 \times). FITC-dextran (10,000 Da) permeabilization after 3 h or 24 h injection of IAPP_s was used to indicate APEL occurrence in mice. FITC-dextran was injected 30 min before the measurement. As expected, there was no obvious vascular leakage observed at 3 h and 24 h with the administration of IAPP_s, consistent with our *in vitro* results (**Figure 2**).

7) Lastly, A β 42 appears to induce EL specifically in the brain and diaphragm. However, the *in vitro* EL effects appear general/non-discriminatory. Does A β 42 have additional preference for EL *in vivo*? Do authors know if A β 42 they inject gets deposited in the brain and diaphragm, as against other organs they have tested for?

Thank you for this question. In the *in vitro* EL experiments, it was difficult to conclude that EL was general or non-discriminatory because only endothelial cells were present. In the *in vivo* experiments, different concentrations of A β species were injected via the tail vein to general circulatory system of the non-transgenic mice. At different concentrations of A β_o /A β_s , APEL appeared in other organs beyond brain and diaphragm. While the anatomy of mice entails drastically different conditions and biological barriers for the transport and localised accumulation of amyloid protein aggregates, at this stage we cannot yet infer a definitive deposition preference for A β based on our data.

Minor concerns

1) What is the justification for using only male mice for experimentation using the A β 42 injection and subsequent measurements involving EBD?

Thanks for your comment. We have supplemented the results for 3 female mice per group in **Figure S14** to complement our data with 3 male mice per group shown in **Figure 6C**. No significant differences were observed in the two experiments.

2) The authors report in vivo results as n=3 independent mice, are these from a single experiment? If so, they need to biologically replicate their experiments for reproducible results and reliable statistical measures.

Thank you for your suggestion. We have used 6 mice for each group in the measurement of vascular leakiness induced by A β _o and A β _s. The data are shown in **Figures 6C&S14**.

Reviewer #2 (Remarks to the Author):

Amyloid formation is a key phenomenon underlying neurodegenerative diseases such as Alzheimer's disease (AD). Understanding the pathogenic mechanisms of the amyloid formation process is important for illustrating the complex causes of AD and could aid the development of new therapeutic strategies.

Li et al. found that amyloid protein oligomers and fibril seeds can lead to endothelial leakiness (APEL), analogous to nanomaterials-induced endothelial leakiness (NanoEL). The size and anionic charge properties of A β aggregates are important for APEL. This process involves extracellular interactions between A β aggregates and vascular endothelial (VE)-cadherin, which triggers a cascade of signaling events and actin remodeling, leading to endothelial leakiness. The authors validated APEL in ex vivo blood vessels and in vivo blood-brain barrier models. Using discrete molecular dynamics simulation tools, the authors characterized some potential binding and functional mechanisms between A β aggregates and VE-cadherin. This work is interesting and has some novel insights into the role of A β aggregates in AD-related vascular permeation. However, there are some concerns that the authors need to address:

We thank the reviewer for this strong endorsement of our work and this comprehensive summary.

Major concerns:

1) In Figure 2, the authors showed that A β oligomers, protofibrils, and seeds could lead to endothelial leakiness, while the monomer and fibril forms of A β cannot. The authors expanded this conclusion to other anionic amyloid proteins like alpha-synuclein, islet amyloid polypeptide, and FapC. However, though the fibril forms of these three proteins were prepared in Figure 1, they were not tested in Figure 2. Without showing the result of their fibril forms, i.e., α Sf, IAPPf, and FapCf, how can the author conclude 'their soluble monomers and lengthy amyloid fibrils were unable to induce leakiness'?

Thank you for this comment. Following this suggestion, we have performed confocal and transwell assays to test endothelial leakiness induced by anionic α S and cationic IAPP with all four species, the monomers, oligomers, fibrils and seeds. The data are updated in **Figures 2&S2**. Unlike the pathogenic A β and α S, FapC is a functional protein whose amyloid form is a fixed anatomical feature of bacteria *Pseudomonas aeruginosa*. The oligomeric and protofibril forms

of functional proteins (including FapC) have rarely been studied as they are often rendered under non-physiological conditions (Ke et al. *Chem. Soc. Rev.* 49, 5473, 2020), and such forms are less likely to be present in the human body compared to fragmented FapCs, shed off bacteria. Therefore we have only used FapC seeds (FapCs) as a minor sidenote to see whether EL could occur beyond the pathogenic triggers of A β and α S. We have modified our statements to be more accurate and specific about A β and α S, on p7&8.

2) There have been many studies regarding the endothelial toxic effects of A β aggregates. The authors found that the APEL process is unique because it does not increase ROS and apoptosis. The reason could be short-term (30 mins) treatment. The result is very interesting. However, the impact of A β aggregates is usually long-term in physiological conditions. It could be possible that APEL is an earlier event before the long-term toxic effect. However, most (if not all) of the cell-based experiments in the study are a short-term treatment of A β aggregates (30mins to 2 hours) and do not extend to long-term exposure, for example, 24-72 hours that other researchers typically use. Therefore, it is hard to discuss the relationship between the short-term and long-term effects. For example, does the long-term effect found in previous reports also involve VE-cadherin signaling events? Can correcting APEL at the early stage reduce long-term toxicity, such as ROS production or apoptosis? A better understanding of the relationship between APEL and the previously reported model(s) could strengthen the novelty of APEL.

Thank you for this very interesting question and observation. Indeed, amyloidogenesis is associated with damage to cells and organs, including the vasculature, through ROS production, inflammation, and gene dysregulation, etc., observed typically on the timescales of 24 hours or longer (Ke et al. *Chem. Soc. Rev.* 46, 6492, 2017). We have made a discussion about this on p10 in the main text. The focus of this study, in comparison, is to reveal a significant new phenomenon which occurs on the timescale of ~an hour *in vitro*, where certain amyloid protein oligomers and seeds engaged with VE-cadherin junctions to cause endothelial leakiness. Such phenomenon is biophysical-biochemical in nature and is independent from toxicological events, as revealed by our signaling pathway assays and simulations. With that said, we do not exclude the possibility that the occurrence of APEL could induce long-term effects either functionally or pathogenically, but such complex topic itself deserves multiple future studies and is beyond the focus of the current study. Indeed, we anticipate that the findings in this study are important for improving our current understanding of the complex and often perplexing roles of amyloidogenesis in connection with vasculature damage and the physiopathology of AD, a viewpoint also seconded by the opening summaries of the three reviewers' reports.

3) In lines 321-322, the author stated, 'Although adding BV-6 may also lead to some significant gap formation in HMVEC monolayers, it only occurred at a much higher BV-6 concentration (dilution of 1:20)'. However, in Figure 4B, the gap area of BV-6 treatment at 1:100 concentration is also significantly higher than media control. Please explain this result that conflicts with the above description. Furthermore, as BV-6 and A β_o /A β_s all lead to a higher gap area compared to media control, could it be possible that the decrease of gap area observed in

(BV-6 1:100 + A β_o) condition is due to the possible interactions between BV-6 and A β aggregates rather than through the VE-cadherin pathway?

Thanks for this comment. We have reorganized the description for the competition assay with BV-6 treatment. As the BV-6 antibody increased in concentration, the VE-cadherin junctions became unstable to gradually break apart to render paracellular gaps, especially at the 1:20 BV-6 dilution. Therefore, it was inaccurate to evaluate the blocking effect at such high antibody concentrations. Meanwhile, the percentage of gap area formation was significantly reduced at the dilutions of 1:100 and 1:500 pre-treatment (BV-6+A β_o /A β_s) compared to A β_o /A β_s only, suggesting the interaction between A β_o /A β_s and VE-cadherin could be blocked by BV-6.

For the procedure of the blocking assay, we first added BV-6 for 1 h incubation, then removed excess BV-6 to minimize the possibility of interaction between BV-6 and A β aggregates. We also performed post-addition of BV-6 after 30 min incubation with A β aggregates. The gap area percentages of the A β_o /A β_s only and the post-addition of BV-6 (A β_o /A β_s +BV-6) yielded no obvious differences (**Figure S8**). This suggests that there was little or no interactions between anti-VEcadherin antibody BV-6 with A β_o /A β_s . Apart from this blocking result, we also tested interactions between VE-cadherin with four different species of A β (A β_m , A β_o , A β_f and A β_s) via a co-immunoprecipitation assay, as presented in **Figure 4E**. We did observe interactions of VE-cadherin with A β_o and A β_s . Taken together, BV-6 could block APEL by first binding to VE-cadherin, thus sterically blocking A β_o /A β_s binding to VE-cadherin.

4) For the immunoprecipitation assay described in lines 338-342, the authors demonstrated the physical binding between A β_o and VE-cadherin. Did the authors compare the binding of VE-cadherin to A β_m , A β_s , and A β_f using the immunoprecipitation assay? The comparison would be necessary to test whether VE-cadherin is essential for different APEL phenotypes triggered by different A β aggregation forms. Furthermore, the results may help to support/compare the *in-silico* modeling observations.

Thank you for this suggestion. In additional co-immunoprecipitation assays, we have probed VE-cadherin interacting with the four different conformations of A β (A β_m , A β_o , A β_f and A β_s), as presented in **Figure 4E**.

We observed interactions with VE-cadherin for A β_o and A β_s , but no detectable interactions for A β_m and A β_f . These results affirmed involvement of VE-cadherin binding to A β_o and A β_s under the experimental conditions. On the other hand, A β_m and A β_f were unable to significantly interact with VE-cadherin, likely due to their softness (monomers) or their stiffer long fibrils. Accordingly, we have added a small discussion on p8&16.

Although A β_m was found to be able to bind VE-cadherin also *in silico*, it did not destabilize VE-cadherin dimer in contrast to A β_o and A β_s . Without disrupting the adherens junctions and opening the intercellular gaps to expose more VE-cadherins for A β_m binding, the amount of A β_m pulled down with VE-cadherins would be undetectable.

5) The authors used APP/PS1 mice or Swiss mice injected with A β_s or A β_o to validate the endothelial leakiness *in vivo*. However, whether or not the observed leakiness is through the VE-cadherin pathway has not been tested. BV-6 antibody or Src-kinase inhibitor, PP1, can be

used to check whether the endothelial leakiness phenomenon can be reversed *in vivo*. These experiments can indicate more clearly the novelty that sets this work different from other studies using evidence from both *in vitro* and *in vivo*. These experiments can also highlight the therapeutic potential of their discovery.

Thank you for this suggestion. Indeed, there could be a therapeutic potential for the APEL discovery. We have performed the suggested experiment *in vivo* accordingly. Src-kinase inhibitor, PP1, was injected through the tail vein at the dose of 1.5 mg/kg/mouse. The assay was designed with 4 groups, including PP1, A β_s , PP1 (1 h) + A β_s , and A β_s (3 h) + PP1. PP1 (1 h) + A β_s refers to pre-injection of PP1 1 h before A β_s administration, and A β_s (3 h) + PP1 refers to 3 h-administration of A β_s followed by PP1 (**Figure 6F, G**).

EBD fluorescence of the brain was measured at 24 h-administration of A β_s . EBD was mixed and injected with A β_s simultaneously. Significant inhibition of APEL was observed at the injection of PP1 after 3 h-administration of A β_s (A β_s (3 h) + PP1) of 2 and 20 μ g compared with A β_s only, suggesting that vascular leakiness induced by A β_s was mediated through the VE-cadherin signalling pathway and APEL indeed occurred but could be reversed *in vivo*. This result was consistent with the *in vitro* data, where PP1 pre-treatment resulted in a significant reduction of leakiness under the treatment of either of the two A β species (**Figure 5**).

6) For the discrete molecular dynamic simulation, the structure of one molecule of A β_m , A β_o , or A β_s was used for modeling the interaction with an EC1 cadherin dimer. How do the authors know the binding ratio between A β species and EC1 cadherin dimer is 1:1? Would the results be different if using two or more molecules of A β species for the modeling?

We agree with the reviewer that the molecular ratios between A β species and EC1 cadherin dimer can be diverse in experiments, related to the aggregation propensity of the A β species as well as the architecture of adherens junction to accommodate the protein aggregates. In addition, due to the heterogeneity and fluctuation in local protein concentration, the molecular ratios could also vary from place to place and over time *in vitro* and *in vivo*. Our atomistic DMD simulations were not intended to reproduce all the experimental observations but to offer additional molecular insights to better understand the binding between the A β species and VE-cadherin as well as disruption to the cadherin dimers by the A β species. Considering the high computational cost, we chose the simplistic molar ratio of 1:1 for our atomistic DMD simulations and we do expect an accumulative/additive effect of such interactions due to availability of multiple binding sites in the dimer (e.g. **Figure 7D**). Accordingly, we have added a brief discussion on page 28.

7) Does the simulation study explain the requirement of anionic charges for amyloid protein to trigger endothelial leakiness? A β_m , A β_o , and A β_s all show different binding mechanisms in the simulation results. A β_o shows strong binding properties to the cationic turn regions of the EC1 dimer, which may explain the need for anionic charge in A β species. However, A β_s that also triggers endothelial leakiness shows a different binding pattern. The authors illustrated the different binding frequencies around the EC1 dimer surface (Fig.7D) but did not describe the binding properties on the structure of A β species to explain the earlier experimental results.

Thank you for the comment. Since cell membranes are anionic, small anionic particulates were able to enter the paracellular space to trigger EL, instead of adsorbing onto cell membranes to evoke endocytosis (Tee *et al. Chem. Soc. Rev.* **48**, 5381, 2019). Here, our simulations focused on the binding of anionic A β species with VE-cadherin already in the paracellular space.

As noted by the reviewer, different A β species displayed different binding properties to the EC1 dimer. A β_0 bound to the cationic turns of the EC1 and triggered the dimer disruption allosterically. As suggested by the reviewer, we have added discussions on p24 and in **Figures S20&S21** about the binding of EC1 with other A β species – i.e., A β_m and A β_s in **Figure 7D**.

8) In the simulation study, the author found that A β_0 significantly increases cadherin dimer's dissociation, while A β_m has a negligible impact on the dimer dissociation. The author suggests that the observation agreed well with the experimental data. However, is cadherin dimer's dissociation or stability involved in the APEL process? There is no direct experimental evidence in the manuscript supporting this.

We apologize for the missing logic gap in the referred discussion. In our prior study (Lee *et al. Adv. Sci.* **8**, e2102519, 2021), we showed that reduced dimer stability and increased dissociation of VE-cadherin resulted in a lower critical tensile force to induce intercellular gap stabilized by a cluster of VE-cadherins. Constantly experiencing mechanical stretches dynamically in living endothelial cell membrane, the adherens junctions with compromised VE-cadherins opens and forms gaps readily. Therefore, our simulation results are consistent with experimental evidence. We have added the above discussion on page 27.

Minor points:

1) Line 195, 'protofirbils' should be 'protofibrils'.

Thanks, we have corrected it accordingly.

2) Line 300, the figure legend indicates 'green: actin', but there is no green in Figure 4A.

Thanks, we have corrected it in the figure legend.

3) It seems that it is better to describe result section 2.7 before result section 2.6 if some of the in-silico modeling results can be validated using ex vivo or in vivo models.

Thanks for your suggestion. The *in silico* results served to complement the *in vitro* and *in vivo* data and provide additional molecular details on APEL, and hence were presented after the experimental work to indicate that reasoning.

Reviewer #3 (Remarks to the Author):

Li and colleagues suggest an attractive hypothesis regarding the roles of amyloid β (A β) aggregates in the pathogenesis of Alzheimer's disease (AD) that extracellular interactions between A β aggregates and VE-cadherin are responsible for the endothelial leakiness, eventually causing pathogenesis of AD. The authors conducted this research not only on experimental methodology, including *in vitro*, *ex vivo*, and *in vivo* assays, but also in *silico* methodology to clarify the mechanism of endothelial leakiness caused by aggregates of A β and several amyloidogenic proteins. I firmly believe that this manuscript attracts broad-spectrum readers because of its importance in the AD pathogenesis. However, several concerns have arisen after my review. The authors should address five points as follows.

We thank the reviewer for this strong endorsement of the significance of our work.

1. Throughout the manuscript, the authors observed endothelial leakiness with aggregates formed using protein solutions with high concentrations, mainly 20-40 μ M, compared with its biological concentrations. Can authors rationalize that the mechanism regarding endothelial leakiness in assays performed *in vitro* can be the same as *in vivo* ones? The authors should clarify the difference in the concentrations of protein between *in vitro* assays and *in vivo* environments and discuss the possible discrepancies in the mechanisms of endothelial leakiness, if any.

AD pathology takes decades to evolve in human. In AD mouse models, such process takes roughly 4-12 months. For practicality, *in vitro* assays typically use high A β concentrations to bring that timescale down to tens of hours. Regardless, the occurrence of APEL is a collective event referring to the rupture of a large number of VE-cadherin pairs upon their exposure to amyloid protein oligomers and seeds. In principle, the opening of a single VE-cadherin pair only requires pN forces (Lee *et al. Adv. Sci.* **8**, e2102519, 2021) and a small number of aggregates, while the concentrations of amyloid beta aggregates much exceed a certain level for APEL to become observable microscopically.

The purpose of this study is to demonstrate the capacity and signalling pathway of anionic amyloid protein aggregates in creating endothelial leakiness, and we have adopted different amyloid protein concentrations to accommodate the timescales and resolution of the *in vitro* and *in vivo* methodologies. We showed that, for all the amyloid protein concentrations applied (1.6 to 20 μ M *in vitro* and 0.002 to 100 μ g per mouse *in vivo*) we consistently observed the occurrence of APEL, and we further revealed the underlining mechanism and characteristics of such novel event. These selected protein concentrations for *in vitro* followed the convention in the field of amyloidogenesis (Ke *et al. Chem. Soc. Rev.* **46**, 6492, 2017). The doses administered to mice spanned five orders of magnitude in range to reflect both the physiological and pathogenic amyloid protein concentrations *in vivo* (Roberts *et al. Brain* **140**, 1486, 2017) and to ensure observations of APEL by fluorescence microscopy and complementary assays.

We have added these justifications to the captions of **Figures 2&6** in the revised manuscript.

2. Throughout the manuscript, although the authors insist that interactions between A β aggregates and VE-cadherin are important in the mechanism on which protein aggregates cause endothelial leakiness, the data regarding the dissociation constant of the interaction are not shown in the manuscript. Do the interactions happen in a specific or nonspecific manner? I understand that the authors tried to reveal this point using a BV-6 antibody (Figure 4B). But adding the BV-6 antibody itself induced endothelial leakiness (the orange bar in Figure 4B), raising the question about the specificity of APEL. Authors should clarify the dissociation constant of the interaction, preferably with experimental data of interaction analyses. Otherwise, readers cannot understand the specificity of the endothelial leakiness caused by protein aggregates.

The importance of VE-cadherin in driving endothelial leakiness induced by anionic nanoparticles has already been established in multiple earlier studies (Setyawati *et al. Nat. Commun.* **4**, 1673, 2013; Tee *et al. Chem. Soc. Rev.* **48**, 5381, 2019; Lee *et al. Adv. Sci.* **8**, e2102519, 2021). Although BV-6 itself induced endothelial leakiness, this observation did not contradict but rather support the critical role of VE-cadherin in inducing APEL. We have also provided blocking assays using a different antibody from BV-6, EC1a, to corroborate our findings on the role of VE-cadherin in APEL (**Figures 4&S7**). The observations that EL can be elicited by anionic nanoparticles in the literature and by amyloid aggregates of both A β and α S as shown by this study suggest that disruptions to VE-cadherin are likely not driven by highly specific interactions but common hydrophobic and charge interactions. We have added a discussion on p24 and included the dissociation rates of cadherin under different forces in the presence of different A β species in **Table S4**.

3. In Figure 1B, the authors defined several species of aggregates on their aggregation pathways. But the definition is ambiguous. For example, what is the difference between A β _o, A β _{o-p1}, and A β _{o-p2}? I recommend the authors clarify the clear differences between identical species.

Thanks for your comment, we agree with reviewer that the definition of A β _{o-p1/2} was ambiguous. We have used A β _{o-p} in the revision and noted on p7 of the revised manuscript that this represented early-stage protofibrils transitioning from the oligomers.

4. In Figure 1G, the authors prepared four kinds of species, monomer (m), oligomer (o), fibrils (f), and seeds (s), for three kinds of proteins, A β , α S, and IAPP. However, the data showing a gap area formed by identical aggregates (Fig. 2A, B, and C) do not include fibril species regarding α S and IAPP. The authors should show the data of these two species (α S_f and IAPP_f) to generalize the hypothesis that the breakdown of mature fibrils into shorter seeds is important to induce endothelial leakiness.

Thank you for this suggestion. We have performed the series assays with the fibrils of A β , α S, and IAPP. The data have been updated in **Figures 2&S2**.

5. In section 2.6 (Fig. 6), the authors suddenly increased the concentration of A β species. The authors should clarify the reason why they increased the concentrations of proteins. Furthermore, the morphologies of aggregates highly depend on solution conditions (i.e., initial

concentrations of proteins). I am afraid that A β 1-42 peptide would form amorphous globular aggregates with concentrations on the order of 100 μ M. Thus, the authors should confirm the characteristics of aggregates formed in higher concentrations of protein monomers.

Thanks for this question. We initially used high protein concentrations *ex vivo* to test the capability of A β species in inducing EL, and then realized that those concentrations were quite high before we switched to much lower concentrations for the *in vivo* assays. The *ex vivo* data have now been removed in the revision as they did not add new insights to the comprehensive results already presented.

REVIEWER COMMENTS

Reviewer #1 (Remarks to the Author):

The authors have performed commendably to address the concerns raised in the previous version. I am satisfied by the additional experimentation that have elevated the specificity of the observations reported in this manuscript and largely in agreement with the justifications provided by the authors in their rebuttal. However, there are two outstanding issues that need to be addressed for the precise interpretation and reproducibility of the data presented in this otherwise well revised version –

1) From Fig.S12, authors conclude that the changes they observe in the APP1/PS1 mice (young vs old) is not observed in the control C57BL/6J mice. If indeed the APP1/PS1 mice is a sound good model for this interpretation, how come the control mice display ~2-3 times higher A β 42 basal levels to begin with across several organs? The only sample where the difference stands out and is consistent with authors interpretation is that of blood. Authors must interpret and discuss these results carefully, rather than merely generalizing the observations from control mice as having no statistically significant pattern as compared to APP1/PS1 mice.

2) For the in vivo EL measurements based on EBD and Dextran assays, are the authors reporting intensity as integrated or mean measurements? How are they normalizing for the brain size, and do they blank for intensity using no EBD or dextran control brains to exclude basal fluorescence from the tissue itself? This aspect is unclear in their legends or the methods section. This information is critical because across Figs. 6, S14, S18, and S19, the striking differences observed in the representative intensity maps of excised brains between treatments are not reflected in the intensity measurements reported in the graphs. In fact, in several brain samples, such as controls or PP1 treated for example, there are barely any specific fluorescence signals detected based on the intensity maps. However, in the graphs, these conditions have a high value and the differences observed between treatments in the intensity is barely reflected in the values presented in the graphs. Is this because there is high basal fluorescence from the tissue that is not being blanked/accounted for? Or are the intensity maps not correctly scaled to appropriately represent the values?

Reviewer #2 (Remarks to the Author):

The authors have addressed my concerns very well. I recommend publication after addressing some minor issues listed below:

- 1) Line 134, What is the 'pI value' for FapC? The pI values for other proteins, such as A β , α -synuclein, and IAPP, were all given, but I didn't see the pI value for FapC?
- 2) The subtitle of 2.3 is 'In vitro APEL occurs independently of A β toxicity or endocytosis'. I think it would be better to use more specific wording such as 'ROS generation, apoptosis, or endocytosis'. 'A β toxicity' is vague - could 'APEL' also be considered as a type of toxic effect of A β ?
- 3) Regarding the experiment shown in Fig. 6A-B, did the authors use 2- and 12-month-old C57BL/6J mice as controls for the EBD injection? In other words, could the increases of leaked EBD be due to aging rather than the changed A β 42 levels?

Reviewer #3 (Remarks to the Author):

Li and coworkers thoroughly addressed the raised concerns that I suggested. Although I give my additional comments as follows, they are minor. The following numbers correspond to the heading numbers of the comments in the previous review. I believe that this manuscript is suitable for publication from Nature Communications after minor revisions.

1. Regarding the concentration of Ab species used in the experiments, the authors gave a reasonable rationalization by quoting appropriate papers. However, the added sentences in the revised manuscript should be placed in the main body of the paper, not in the captions.
 3. Although the authors changed the terminology for each amyloid species, the difference between oligomers and protofibrils remains ambiguous. Authors should clarify the difference between these species based on some data. For a good example, Miti and colleagues clearly distinguished several aggregation species, such as oligomer, rigid fibrils, and curvilinear fibrils, based on the optical measurements (T. Miti et al., *Biomacromolecules*, 16, 326-335 (2015)). I guess the difference between these species in this study is their apparent size, not their secondary structure. Such an ambiguous definition makes readers confused.
- 2, 4, and 5. I confirmed the revision and have no comments.

We thank the three reviewers for their unanimous and strong endorsements of our revised manuscript. In this second revision, we have fully addressed the remaining questions by the reviewers to further strengthen the presentation of our manuscript. All changes are highlighted in red for easy viewing in the second revision. We sincerely thank the reviewers for their insights, efforts and time and our appreciation was conveyed in the Acknowledgement of the manuscript.

Reviewer #1 (Remarks to the Author):

The authors have performed commendably to address the concerns raised in the previous version. I am satisfied by the additional experimentation that have elevated the specificity of the observations reported in this manuscript and largely in agreement with the justifications provided by the authors in their rebuttal. However, there are two outstanding issues that need to be addressed for the precise interpretation and reproducibility of the data presented in this otherwise well revised version –

We thank the reviewer for the endorsement of our revision.

1) From Fig.S12, authors conclude that the changes they observe in the APP1/PS1 mice (young vs old) is not observed in the control C57BL/6J mice. If indeed the APP1/PS1 mice is a sound good model for this interpretation, how come the control mice display ~2-3 times higher A β 42 basal levels to begin with across several organs? The only sample where the difference stands out and is consistent with authors interpretation is that of blood. Authors must interpret and discuss these results carefully, rather than merely generalizing the observations from control mice as having no statistically significant pattern as compared to APP1/PS1 mice.

We thank the reviewer for identifying this human error which occurred when we were converting the units from ng/mL to ng/mg for the A β levels in Fig. S12C. We have corrected and updated the figure.

2) For the in vivo EL measurements based on EBD and Dextran assays, are the authors reporting intensity as integrated or mean measurements? How are they normalizing for the brain size, and do they blank for intensity using no EBD or dextran control brains to exclude basal fluorescence from the tissue itself? This aspect is unclear in their legends or the methods section. This information is critical because across Figs. 6, S14, S18, and S19, the striking differences observed in the representative intensity maps of excised brains between treatments are not reflected in the intensity measurements reported in the graphs. In fact, in several brain samples, such as controls or PP1 treated for example, there are barely any specific fluorescence signals detected based on the intensity maps. However, in the graphs, these conditions have a high value and the differences observed between treatments in the intensity is barely reflected in the values presented in the graphs. Is this because there is high basal fluorescence from the tissue that is not being blanked/accounted for? Or are the intensity maps not correctly scaled to appropriately represent the values?

Thank you for this comment. The missing information has now been added to the Method section. For the in vivo EL measurements, we used mean signal intensity

(photons/second/cm²/steradian) to represent the EBD or FITC-dextran levels in the brain. The signal areas of the brain were also calculated by the software automatically. We have used the no EBD or dextran control to blank the basal fluorescence of tissue itself during the experiments, such as the figure below, where the right panel depicts the brain of a mouse without EBD injection. The fluorescence intensity of the non-EBD injected mouse brain had similar values with the groups which had EBD injection. For the control groups in our study, the mice also received the same amount of EBD or FITC-dextran as the sample groups, and the fluorescence intensities of the sample groups in the graph were normalized to the corresponding mean values from the control group. To better observe the differences between the groups in the images, the scales of the intensity maps in Figs. 6, S14, S18, and S19 were adjusted, but this did not result in any data value changes. We have updated the figures with the restored scales for Figs. 6, S15, S19, and S20.

Figure 1. No EBD control was used to blank the basal fluorescence of tissue itself during the experiments. Left panel corresponds to **Figure S14A**, where the mice received EBD injection, and were then imaged after 24 h. Right panel depicts the brain of a mouse without EBD injection.

Reviewer #2 (Remarks to the Author):

The authors have addressed my concerns very well. I recommend publication after addressing some minor issues listed below:

We thank the reviewer for the strong endorsement of our revision.

1) Line 134, What is the 'pI value' for FapC? The pI values for other proteins, such as A β , α -synuclein, and IAPP, were all given, but I didn't see the pI value for FapC?

Thanks for this comment. According to our previous publication, FapC monomers are near charge neutral, but FapC amyloid fibrils are anionic (zeta potential: -36 mV).¹ We have cited this zeta potential value in the revised manuscript as we used FapC seeds derived from FapC amyloid fibrils.

2) The subtitle of 2.3 is 'In vitro APEL occurs independently of A β toxicity or endocytosis'. I think it would be better to use more specific wording such as 'ROS generation, apoptosis, or endocytosis'. 'A β toxicity' is vague - could 'APEL' also be considered as a type of toxic effect of A β ?

Thank you for this suggestion. We have modified the subtitle accordingly.

3) Regarding the experiment shown in Fig. 6A-B, did the authors use 2- and 12-month-old C57BL/6J mice as controls for the EBD injection? In other words, could the increases of leaked EBD be due to aging rather than the changed A β ₄₂ levels?

Thanks for this comment. We have compared the leakiness across the blood-brain barrier through a new measurement of EBD permeabilization 24 h post-injection in background strain C57BL/6J mice at 2- and 12-months old (Fig. S14). There were no significant changes of the EBD intensities between the 2- and 12-months old mice, indicating that aging alone without the onset of AD in the background mice did not affect the integrity of their blood-brain barrier. Therefore, the increases of leaked EBD in the mice of our study were due to the changing A β ₄₂ levels.

Reviewer #3 (Remarks to the Author):

Li and coworkers thoroughly addressed the raised concerns that I suggested. Although I give my additional comments as follows, they are minor. The following numbers correspond to the heading numbers of the comments in the previous review. I believe that this manuscript is suitable for publication from Nature Communications after minor revisions.

We thank the reviewer for the strong endorsement of our revision.

1. Regarding the concentration of Ab species used in the experiments, the authors gave a reasonable rationalization by quoting appropriate papers. However, the added sentences in the revised manuscript should be placed in the main body of the paper, not in the captions.

Thanks for the suggestion, we have updated the text accordingly, on p8, highlighted in red.

3. Although the authors changed the terminology for each amyloid species, the difference between oligomers and protofibrils remains ambiguous. Authors should clarify the difference between these species based on some data. For a good example, Miti and colleagues clearly distinguished several aggregation species, such as oligomer, rigid fibrils, and curvilinear fibrils, based on the optical measurements (T. Miti et al., *Biomacromolecules*, 16, 326-335 (2015)). I guess the difference between these species in this study is their apparent size, not their secondary structure. Such an ambiguous definition makes readers confused.

We thank the reviewer for this suggestion. In our study, we based on incubation time (age) and morphology by TEM for assigning the terminologies of the oligomers and the protofibrils. As mentioned in the literature,² the very early species on path of amyloid formation are found to be dimers, trimers, tetramers, collectively termed as oligomers. Oligomers may have variable sizes, with an average of approximately 25-30 protein molecules, ranging in size from 20 to >50 kDa and further addition of monomeric unit to oligomers can result in the formation of bead-like structures up to 200 nm in length called protofibrils. We have modified the text and cited this quoted reference in our revised manuscript, on p7.

2, 4, and 5. I confirmed the revision and have no comments.

Thanks for the comment.

References

1. Huma, Z. E., *et al.* Nanosilver mitigates biofilm formation via fapc amyloidosis inhibition. *Small* **16**, e1906674 (2020).
2. Siddiqi, M. K., Majid, N., Malik, S., Alam, P., Khan, R. H. Amyloid oligomers, protofibrils and fibrils. In: *Macromolecular protein complexes ii: Structure and function* (eds Harris JR, Marles-Wright J). Springer International Publishing (2019).